# Effect of mid-term drought on *Quercus pubescens* BVOCs emissions seasonality and their dependence to light and/or temperature

Amélie Saunier[1], Elena Ormeño[1], Christophe Boissard[2], Henri Wortham[3], Brice Temime-Roussel[3], Caroline Lecareux[1], Alexandre Armengaud[4], Catherine Fernandez[1].

[1]Aix Marseille Univ, Univ Avignon, CNRS, IRD, IMBE, Marseille, France.

[2]Laboratoire des Sciences du Climat et de l'Environnement, LSCE/IPSL, CEA-CNRS-UVSQ, Université Paris-Saclay, F-91191 Gif-sur-Yvette, France.

[3]Aix Marseille Univ, CNRS, LCE, Laboratoire de Chimie de l'Environnement, Marseille, France

[4] Air PACA, 146 rue Paradis, Bâtiment Le Noilly Paradis, 13294 Marseille, Cedex 06.

*Correspondence to:* Amélie Saunier (amelie.saunier@imbe.fr)

**Key words**: BVOCS, natural and amplified drought, season, light and temperature

**Abstract.** Biogenic volatile organic compounds (BVOCs) emitted by plants represent a large source of carbon compounds released into the atmosphere where they account for precursors of tropospheric ozone and secondary organic aerosols. Being directly involved in air pollution and indirectly in climate change, understanding what factors drive BVOC emissions is a prerequisite for modelling their emissions and predict air pollution. The main algorithms currently used to model BVOCs emissions are mainly light and/or temperature dependent. Additional factors such as seasonality and drought also influence isoprene emissions, especially in the Mediterranean region which is characterized by a rather long drought period in summer. These factors are increasingly included in models but only for the principal studied BVOC, namely isoprene but there are still some discrepancies in estimations of emissions. In this study, the main BVOCs emitted by *Quercus pubescens*: isoprene, methanol, acetone, acetaldehyde, formaldehyde, MACR, MVK and ISOPOOH (these 3 last compounds detected under the same m/z), were monitored with a PTR-ToF-MS over an entire seasonal cycle, under both *in situ* natural and amplified drought which is expected with climate change. Amplified drought impacted all studied BVOCs by reducing emissions in spring and summer while increasing emissions in autumn. All six BVOCs monitored showed daytime light and temperature dependencies while three BVOCs (methanol, acetone and formaldehyde) also showed emissions during the night despite the absence of light under constant temperature. Moreover, methanol and acetaldehyde burst in the early morning and formaldehyde deposition/uptake were also punctually observed which were not assessed by the classical temperature and light models.

## 1 Introduction

Plants contribute to global emissions of volatile organic compounds (VOCs) with an estimated emission rate of $10^{15}$ gC yr$^{-1}$ (Guenther *et al.* 1995; Harrison *et al.* 2013). The large variety of compounds released by plants represents, at the global scale, 2-3% of the total carbon released in the atmosphere (Kesselmeier & Staudt 1999). Under strong photochemical conditions, BVOCs, together with NO$_x$, can significantly contribute to tropospheric ozone concentration (Xie *et al.* 2008; Papiez *et al.* 2009). In addition to its greenhouse effect, O$_3$ has strong effects on plant metabolism (Reig-Armiñana *et al.* 2004; Beauchamp *et al.* 2005) as well as on human health (Lippmann 1989). BVOCs are also rapidly oxidized by OH radical and NO$_3$ (Hallquist *et al.* 2009; Liu *et al.* 2012), which account for an important fraction of the total mass of secondary organic aerosols (SOA, Jimenez *et al.* 2009). Methanol and acetone are, after isoprene, the principal BVOC released to the atmosphere. Isoprene emissions represent between 400-600 TgC yr$^{-1}$ at the global scale (Arneth *et al.* 2008) whereas methanol emissions vary between 75 and 280 TgC yr$^{-1}$ (Singh *et al.* 2000; Heikes *et al.* 2002, respectively) and acetone emissions represent only 33 TgC yr$^{-1}$ (Jacob *et al.* 2002). Other compounds such as acetaldehyde, methacrolein (MACR), methyl vinyl ketone (MVK), isoprene hydroxy hydroperoxides (ISOPOOH) and formaldehyde, whose biogenic origin has been poorly investigated, are better known to be anthropogenic and/or secondary VOCs issued from atmospheric oxidations (Hallquist *et al.* 2009). However, acetaldehyde is also a by-product of plant metabolism and its emissions represent 23 Tg yr$^{-1}$ at the global scale (Millet *et al.* 2010). Formaldehyde, MACR, MVK and ISOPOOH are released by plants through oxidations of methanol and isoprene, respectively, within leaves but they can have other leaf precursors (Oikawa & Lerdau 2013). Thus, it is thereby important to model all this panel of BVOCs emissions with the aim of predicting their effect on secondary atmospheric chemistry.

Current models allow to predict BVOCs emissions according to the type of vegetation, biomass density, leaf age, specific emission factor for many vegetal species, as well as the impact of some environmental factors. Models, such as the MEGAN (Guenther *et al.* 2006; Guenther *et al.* 2012) or CHIMERE (Menut *et al.* 2014) model, include at least two main algorithms that allow to model light and temperature emissions dependence (called *L+T* algorithm afterwards) and a temperature dependent algorithm (called *T* algorithm afterwards), both described in Guenther *et al.* (1995). The L+T algorithm is typically used for BVOCs emissions whose synthesis rapidly relies on photosynthesis, and hence include *de novo* emissions. The T algorithm is used for BVOCs emissions that do not directly rely on BVOCs synthesis when, for example, they originate from permanent large storage pools (Ormeno *et al.* 2011). The dependence to light and/or temperature is well documented for isoprenoids (Owen *et al.* 2002; Rinne *et al.* 2002; Dindorf *et al.* 2006) but there is still a lack of knowledge about highly volatile BVOCs (e.g. methanol, acetone, acetaldehyde). However, many of these compounds are very reactive in the atmosphere (Hallquist *et al.* 2009) and, could be emitted in large quantities to the atmosphere at global scale. The characterization of their emissions and sensitivity to light and/or temperature is, thus, necessary in order to obtain reliable predictions of atmospheric processes in order not to miss this important part of the atmospheric reactivity. Other factors than light and temperature can drive BVOCs emissions such as water stress. Most studies dealing with BVOCs response to water stress have, however, focused on terpene-like compounds and have been carried out after weeks of watering restriction or removal under controlled conditions (for a review, see studies cited in Peñuelas and Staudt 2010). Considerable uncertainty remains in our understanding of emission mechanisms since some works showed increases (Funk *et al.* 2004; Monson *et al.* 2007) or decreases of isoprene emissions (Brüggemann & Schnitzler 2002; Fortunati *et al.* 2008) and there is a lack of knowledge on the impact of water stress on highly volatile BVOCs emissions (e.g. methanol). Moreover, the understanding of isoprene sensitivity

and highly volatile BVOCs to recurrent water stress (few years) under *in situ* conditions is clearly missing.
Likewise, the capacity of current $L+T$ and $T$ algorithms to predict emission shifts under different drought scenarios
in the context of climate change needs to be addressed for isoprene and highly volatile compounds. This is of
especial interest for the Mediterranean area where the most severe climatic scenario of the IPCC predicts an
intensification of summer drought consisting on a rain reduction that can locally reach 30%, an extension of the
drought period as well as a temperature rise of 3.4°C, (Giorgi & Lionello 2008; IPCC 2013; Polade *et al.* 2014)
for 2100.
In the present investigation, we aimed (i) to study the emission factors of each studied BVOC released by *Q.*
*pubescens,* including isoprene and highly volatile compounds that originate from plant metabolism under water
stress (ii) to test the performance of the L+T and T algorithms to predict isoprene and highly volatile BVOC
emissions over the seasonal cycle and under two recurrent water stress treatments. *Q. pubescens* was chosen as
vegetal model because this species is highly resistant to drought and well widespread in the Northern
Mediterranean area occupying 2 million ha (Quézel & Médail 2003). It also represents the major source of isoprene
emissions in the Mediterranean area and the second one at the European scale (Keenan *et al.* 2009).
**2 Material and methods**
**2.1 Experimental site**
Our study was performed at the $O_3HP$ site (Oak Observatory at OHP, Observatoire de Haute Provence), located
60 km North of Marseille, France (5°42'44" E, 43°55'54" N), at an elevation of 650m above the sea level. The
$O_3HP$ (955m$^2$), free from direct human disturbance for 70 years, is a homogeneous forest mainly composed of *Q.*
*pubescens* ($\approx$ 90 % of the biomass and $\approx$ 75 % of the trees) with a mean diameter of 1.3 m. The remaining 10 %
of the biomass is mostly represented by *Acer monspessulanum* trees, a very low isoprene-emitter species (Genard-
Zielinski *et al.* 2015). The $O_3HP$ site was created in 2009 in order to study the impact of climate change on a *Q.*
*pubescens* forest. Using a rainfall exclusion device (an automated monitored roof deployed during chosen rain
events) set up over part of the $O_3HP$ canopy, it was possible to reduce natural rain by 30% and to extend the
drought period in an attempt to mimic the current climatic model projections for 2100 (Giorgi & Lionello 2008;
IPCC 2013; Polade et al. 2014). Two plots were considered in the site; a plot receiving natural precipitation where
trees grew under natural drought (300m² surface, used as control plot) and a second plot submitted to amplified
drought (232m² surface). Rain exclusion on this latter plot started on May 2012 and was continuously applied
every year, principally, during the growth period. Ombrothermic diagrams indicate that the drought period was
extended for 2 months in 2012, 4 months in 2013 and 3 months in 2014 for amplified drought relative to natural
drought (Fig 1). Data on cumulative precipitation show that 35% of rain was excluded in 2012 (from 29 April from
to 27 October), 33.5% in 2013 (from 7 July from to 29 December), 35.5% in 2014 (from 8 April to 8 December).
This experimental set up involved a recurrent drought in the amplified drought plot. Sampling was performed at
the branch-scale at the top of the canopy during three campaigns from October 2013 to July 2014, covering an
entire seasonal cycle: in autumn (14 to 28 October 2013, 2[nd] year of amplified drought), in spring (12 to 19 May
2014, 3[rd] year of amplified drought) and in summer (13 to 25 July 2014, 3rd year of amplified drought). Spring,
summer and autumn campaigns corresponded to the end of leaf growth, leaf maturation and the beginning of the
leaf senescence, respectively. The same five trees per plot were selected and investigated throughout the study.

## 2.2 Branch scale-sampling methods

Two identical dynamic branch enclosures were used for sampling gas exchange (in terms of $CO_2$, $H_2O$ and BVOCs) as fully described in Genard-Zielinski *et al.* (2015) with some modifications. Branches were enclosed in a $\approx$ 30L PTFE (polytetrafluoroethylene) frame closed by a 50μm thick PTFE film. One tree from natural and one tree from amplified drought plot were analysed concomitantly during 1 or 2 days. Inlet air was introduced at 9L.min$^{-1}$, controlled by mass flow controllers (MFC, Bronkhorst), using a pump, inside, by PTFE (KNF N840.1.2FT.18®, Germany) allowing for air renewal inside the chamber every ~ 3min. Ozone was removed from inlet air by placing PTFE filters impregnated with sodium thiosulfate ($Na_2S_2O_3$) as described by Pollmann *et al.* (2005), so that oxidation of BVOCs due to ozone within the enclosed atmosphere is negligible. The excess of air humidity was removed using drierite. A PTFE fan ensured a rapid mixing of the chamber air and a slight positive pressure within the enclosure enabled the PTFE film to be held away from leaves to minimise biomass damage. Microclimate (temperature, relative humidity and photosynthetically active radiation or PAR) was continuously (every minute) monitored by a data logger (LI-COR 1400®; Lincoln, NE, USA) with a relative humidity and temperature probe placed inside the chamber (RHT probe, HMP60, Vaisala, Finland) and a quantum sensor (PAR, LI-COR, PAR-SA 190®, Lincoln, NE, USA) placed outside the chamber. The climatic conditions in terms of PAR and temperatures are summarized in Fig. S1 (in supplementary files) for each field campaigns. All air flow rates were controlled by mass flow controllers (MFC, Bronkhorst) and all tubing lines were made of PTFE. Chambers were installed the day before measurements and flushed overnight. Enclosed branches contained 8 to 12 leaves corresponding to a range of 1.4 to 3.6 g of dry matter and 110 to 320 cm² of leaf surface, respectively.

## 2.3 Ecophysiological parameters

Exchange of $CO_2$ and $H_2O$ from the enclosed branches was continuously (every min) measured using infrared gas analysers (IRGA 840A®, LI-COR) concomitantly with BVOCs emission measurements (cf. 2.4). Gas exchange values were averaged by taking into account all the data measured between 12h and 15h (local time). Net photosynthesis ($Pn$, μmol$CO_2$ m$^{-2}$ s$^{-1}$) and stomatal conductance to water ($Gw$, mmol$H_2O$ m$^{-2}$ s$^{-1}$) were calculated using equations described by Von Caemmerer and Farquhar (1981) as used in Genard-Zielinski *et al.* (2015) (for more details, see Appendix A, equations A1 to A4). Leaves from enclosed branches were directly collected after gas exchange sampling to accurately measure leaf surface with a leaf area meter. $Pn$ and $Gw$ were hence expressed in a leaf surface basis. After that, leaves were freeze-dried to assess their dry mass.

## 2.4 BVOCs analysis

A PTR-ToF-MS 8000 instrument (Ionicon Analytik GmbH, Innsbruck, Austria) was used for online measurements of BVOCs emitted by the enclosed branches. A multi-position common outlet flow path selector valve system (Vici) and a vacuum pump were used to sequentially select air samples from: amplified drought, inlet air, natural drought, ambient air and catalyst. The catalyst consists in a 25 cm long stainless steel tubing, filled with platinum wool and heated at 350°C to efficiently remove VOCs from sample and measure potential instrumental background levels. Each sample was analysed every hour, with 15min of analysis. Mass spectra in the range 0-500amu were recorded at 1min integration time. The reaction chamber pressure was fixed at 2.1mbar, the drift tube voltage at 550V and the drift tube temperature at 313 K corresponding to an electric field strength applied to the drift tube

(E) to a buffer gas density (N) ratio of 125Td ($1Td = 10^{-17}$ V cm$^2$). A calibration gas standard, consisting of a
mixture of 14 aromatic organic compounds (TO-14A Aromatic Mix, Restek Corporation, Bellefonte, USA, 100 ±
10ppb in Nitrogen), was used to experimentally determine the ion relative transmission efficiency. BVOCs
targeted in this study and their corresponding ions include formaldehyde (m/z 31.018), methanol (m/z 33.033),
acetaldehyde (m/z 45.03), acetone (m/z 59.05), isoprene (m/z 41.038, 69.069) and MACR+MVK+ISOPOOH (m/z
71.049, these three compounds were detected with the same m/z with PTR-MS). The signal corresponding to
protonated VOCs was converted into mixing ratios by using the proton transfer rate constants k given by Cappellin
*et al.* (2012). Formaldehyde concentrations were calculated according to the method described by Vlasenko *et al.*
(2010) to account for its humidity dependent sensitivity.
BVOCs emissions rates (ER) were calculated by considering the BVOCs concentrations in the inlet and outlet air
as follows (equation 1):
$$ER = \frac{Q_0 * (C_{out} - C_{in})}{B} \qquad\qquad (1)$$
where *ER* was expressed in µgC $g_{DM}^{-1}$ h$^{-1}$, $Q_0$ was the flow rate of the air introduced into the chamber (L h$^{-1}$), $C_{out}$
and $C_{in}$ were the concentrations in the inflowing and outflowing air (µgC L$^{-1}$), respectively, and B was the total
dry biomass matter ($g_{DM}$). Daily cycles were made by averaging measured emissions of all trees every hour.
**2.5 Emission algorithms**
The light and/or temperature dependence of *Q. pubescens* BVOCs (isoprene and highly volatile compounds) under
natural and amplified drought was tested using both the *L+T* and *T* algorithms. Emission rates calculated according
to these algorithms (afterwards, called $ER_{L+T}$ and $ER_T$, respectively) were calculated using the equations described
in Guenther *et al.* (1995) (for more details, see Appendix B, equations B1 to B5). The empirical coefficient β (used
in the *T* algorithm) was determined for each compound according to the season and the treatment through the slope
of correlation between the natural logarithm of emissions rates (measured emissions, µgC $g_{DM}^{-1}$ h$^{-1}$) and
experimental temperature (K). Emissions factors (*EF*), that are emissions rates at standard conditions of light and
temperature, 1000µmol m$^{-2}$ s$^{-1}$ and 30°C), were used to calculate modelled emissions and were determined for each
compound under each season and treatment tree by tree. *EF* values correspond to the slope of the correlation
between experimental emission rates and $C_l*C_t$ when using the *L+T* algorithm or $C_T$ when using the *T* algorithm
(without forcing data to pass through the origin, see Appendix B for a full description of $C_l*C_t$ and $C_T$). R² and p-
value of these correlations tree by tree are presented in tables S1 – S6 (supplementary files) and all parameters
used for the calculation of modelled emissions are presented in tables S7 and S8 (for $Cl*Ct$ and $C_T$, respectively,
in supplementary files).
**2.6 Data treatment**
Data treatment was performed with the software STATGRAPHICS® centurion XV (Statpoint, Inc). After having
checked the normality of the data set, two-way repeated measures ANOVA were carried out to evaluate the
variability of *Pn*, *Gw* and BVOC emission rates according to the drought treatment and season. Correlation
coefficient (R²) and slope (called "sl" afterwards) from Pearson's correlations between measured and modelled
emissions were used to evaluate the algorithm (*L+T* or *T*) that better predicted *Q. pubescens* emissions under the
different drought conditions and seasonal cycle. These correlations indicate if there was an under- or over-
estimation of modelled emissions with sl < 1 and sl > 1, respectively, or if the intercept (called "b" afterwards) are
different from 0. For that, slope comparison tests were performed to check for slope significant differences from
1 and intercept tests were performed to check for intercept significant differences from 0. These correlations were
obtained without forcing data to pass through the origin and with this relation: modelled emissions = a*measured
emission + b.

## 3. Results and discussion

### 3.1 Ecophysiological parameters

The physiology of *Q. pubescens* was slightly impacted by amplified drought over the whole study (Fig. 2), with a
decrease of *Gw* under amplified drought compared to natural drought – ranging from 44 % in spring ($P < 0.1$) to
55 % in summer ($P < 0.01$, Table 1). In autumn, there was no significant difference between both treatments. Pn
was only slightly reduced in summer by 36 % ($P < 0.1$) with no difference for the others season. Thus, the stomatal
closure observed had a slight impact on carbon assimilation. Indeed, *Q. pubescens* has a high stem hydraulic
efficiency (Nardini & Pitt 1999) which compensates stomatal closure since it allows to use water more efficiently,
thus, maintaining *Pn*. Moreover, it must be noted that an increase of *Pn* was observed in autumn and could likely
be attributed to autumnal rains. These results showed that the amplified drought artificially applied to *Q. pubescens*
at $O_3HP$ led to a moderate drought for this species, based on a moderate reduction of the physiological
performances (Niinemets 2010).

### 3.2 Effect of drought on BVOCs emissions

Emissions of all BVOCs followed during this experimentation were reduced under amplified drought compared
to natural drought, especially in spring and summer (Table 1) except for acetaldehyde emissions. Indeed,
acetaldehyde was not significantly different between both treatments probably due to a large variability of the data
set. In autumn, for all BVOCs, there was no difference between both plots. The decrease of oxygenated BVOCs
in spring and summer under amplified drought (e.g. methanol, MACR+MVK+ISOPOOH, formaldehyde, acetone)
could be explained by stomatal closure in spring and summer under amplified drought since emissions of these
compounds are strongly bound to *Gw* (Niinemets *et al.* 2004). Isoprene emissions were also reduced in spring and
summer during the 3[rd] year of this experiment whereas an increase had been observed in the first year (Génard-
Zielinski *et al.* in prep) as well as what had been shown by Brüggemann and Schnitzler (2002) but this work was
conducted with potted plants. The isoprene decrease observed in our experiment cannot be explained by the
stomatal closure because this compound could also be emitted through the cuticle (Sharkey & Yeh 2001). It could
rather be due to the decrease of *Pn* which reduced the carbon availability to produce isoprene. Moreover, carbon
assimilated through *Pn* can be also invested into the synthesis of other defense compounds leading to a decrease
of isoprene production and emission.

### 3.3 Effect of drought on light and/or temperature dependence through a seasonal cycle

All six BVOCs monitored showed daytime light and temperature dependencies (isoprene, degradation products of
isoprene and acetaldehyde), while three BVOCs (methanol, acetone and formaldehyde) also showed emissions
during the night despite the absence of light under constant temperature.

Regarding the light and temperature dependencies, the daily cycle of isoprene emissions (Fig. 3) showed that this
compound clearly responds to light and temperature as already known (Guenther *et al.* 1993) and that this response
is not impacted by amplified drought. Isoprene can protect thylakoids from oxidative damage (Velikova *et al.*
2011) occurring mainly during the day which can explain this kind of dependence. Yet, our results show the
intensity of isoprene emission factor under natural and amplified drought is very different independently of the
season. The modelled emissions were roughly very representative of measured emissions. We note, however, that
in spring, under natural drought, emissions were slightly underestimated (sl = 0.84, $P < 0.05$, $R^2 = 0.90$). It suggests
that although light and temperature remain the main factors driving isoprene emissions in spring but other
parameters explain 10% of these emissions. At this season, plants likely needed to produce more isoprene to protect
the establishment of photosynthetic machinery in the new leaves which could slightly modify the effects of light
and temperature on isoprene emissions.
MACR+MVK+ISOPOOH emissions, as isoprene, seemed to respond better to light and temperature than to only
temperature (Fig. S2 in supplementary files) since correlations between measured emissions and $ER_{L+T}$ were
always better than correlations with $ER_T$. Since MACR+MVK+ISOPOOH are oxidation products of isoprene
(Oikawa & Lerdau 2013), it is not surprising that these compounds followed the same pattern than isoprene in
terms of dependence to light and temperature. The estimations of $ER_{L+T}$ were quite good except in spring under
natural drought where a slight underestimation was observed (sl = 0.87, $P < 0.05$). This underestimation can be
explain by the underestimation of isoprene emissions observed at the same time since MACR+MVK+ISOPOOH
comes from isoprene oxidation.
The dependence of acetaldehyde emissions to light and/or temperature is very contrasted; studies have shown that
they are bound to both light and temperature (Jardine 2008; Fares *et al.* 2011) or to temperature only (Hayward *et*
*al.* 2004). Our results suggested that acetaldehyde emissions were mainly bound to light and temperature (Fig. 4).
Indeed, correlations between measured and $ER_{L+T}$ were always better than with $ER_T$. However, some discrepancies
were observed. Under natural drought, underestimations were observed in spring and summer (sl = 0.72, and sl =
0.57, $P < 0.05$, respectively) whereas in autumn, there was a good estimation (sl = 0.86, $P > 0.05$). Under amplified
drought, underestimation was only observed in summer (sl = 0.80, $P < 0.05$). Trees studied in this experiment did
not show the same dependence to light and temperature for acetaldehyde emissions. $R^2$ of the correlation
determining EF (performed tree by tree), varies from 0.34 to 0.90 in summer, from 0.67 to 0.92 in spring, under
natural drought. Under amplified drought, $R^2$ varies from 0.22 to 0.83 in summer (Tables S6 in supplementary
files). These results suggest that the effect of light and temperature on acetaldehyde emissions strongly depend on
tree considered and could explain the underestimations observed in our experiment. Moreover, daily cycles of
acetaldehyde emissions presented also an emissions burst in the morning (at 7h, local time) in spring (under both
treatments) and in summer (only under natural drought). Acetaldehyde can be produced due to an overflow of
pyruvic acid during light-dark transitions. Cytosolic pyruvic acid levels rise rapidly and it can be converted into
acetaldehyde by pyruvate decarboxylase (Fall 2003). This mechanism could explain the morning burst for this
compound and the fact that no emissions during the night was observed.

We observed emissions of methanol, acetone and formaldehyde during the night under no light and constant temperature (around 20°C, see supplementary files S1). Correlations between $ER_{L+T}$ or $ER_T$ and measured methanol emissions were very similar especially in spring and summer (Fig. 5). However, some observed phenomena suggested that methanol emission was sustained by temperature in the absence of light. Indeed, the burst in the early morning (at 7h, local time), similar to acetaldehyde, was observed when stomata opened in spring and summer, independently of the drought treatment although it was clearer under natural than amplified drought. This burst can be explained by a strong release of this compound that has been accumulated in the intercellular air space and leaf liquid pools (due to the relative high polarity of methanol) at night when stomata are closed (Hüve *et al.* 2007). Moreover, for both drought treatments, methanol emissions during the night were observed at any seasons (especially autumn) which could be explained by nocturnal temperatures (roughly constant) that sufficed to maintain the biochemical processes involved in methanol formation. Methanol emissions, which result from the demethylation of pectin during the leaf elongation, has already been described to be temperature dependent alone (Hayward *et al.* 2004; Folkers *et al.* 2008). However, our results suggest that methanol emissions respond strongly to light and temperature during the day. This kind of diurnal emissions cycle has already been described by Smiatek and Steinbrecher (2006). Our results about daily cycles of acetone emissions (Fig. S3 in supplementary files) showed that this compound responded better to light and temperature than only temperature since correlations were better with $ER_{L+T}$. Under natural drought, the modelled emissions were well representative of measured emissions in summer. By contrast, in spring and in autumn, slight underestimations were observed (sl = 0.88, $P < 0.05$ and sl = 0.69, $P < 0.05$, respectively). Under amplified drought, good estimations were observed in summer and autumn but in spring, there was an overestimation of modelled emissions (sl = 1.27, $P < 0.05$). Previous studies have shown that acetone rather depends on temperature alone (Fares *et al.* 2011) or to light and temperature (Jacob *et al.* 2002), indicating that its dependence on light and/or temperature remains unclear. During the day, acetone emissions were dependent on light and temperature and emissions still occurred during the night, especially in autumn. Alike methanol, nocturnal temperatures could allow to maintain acetone formation (Smiatek & Steinbrecher 2006). Acetone is a by-product of plant metabolism (Jacob *et al.* 2002) and its production can be enzymatic and non-enzymatic (Fall 2003) which can explain these observed differences through the day. We can suppose that acetone emissions observed during the day could come from the enzymatic activity and, on the contrary, during the night, they could come from the non-enzymatic production.

Formaldehyde emissions followed the same pattern than methanol and acetone emissions (Fig. S4 in supplementary files), especially in autumn. By considering only the daytime (correlation with $L+T$ modelled emissions), there were good estimations in summer and autumn and a slight underestimation was observed in spring (sl = 0.89, $P < 0.05$) for natural drought. Under amplified drought, correlations indicated that $L+T$ modelled emissions were well representative of measured emissions, but some negative emissions were observed in summer which suggested a deposition or an uptake of this compound by leaves as already highlighted by Seco *et al.* (2008). This phenomenon could have a role in stress tolerance, since formaldehyde can be catabolised (mainly through oxidations) within leaves leading to $CO_2$ formation (Oikawa & Lerdau 2013). Emissions during the night suggest that formaldehyde came from another source than oxidation within leaves since oxidations occur mainly during the day due to an excess of light in chloroplasts, principal place of reactive oxygen species production (Asada 2006). Thus, formaldehyde emissions observed during the night could result from, for example, the glyoxylate

decarboxylation or the dissociation of 5,10-methylene-THF (Oikawa & Lerdau 2013). Predicting emissions rates
of these 3 compounds (methanol, acetone and formaldehyde), during the night, seem to require other parameters
such as a temperature threshold, below which methanol, acetone and formaldehyde synthesis and so emissions do
not occur.

## 4 Conclusion

After 3 years of amplified drought, all BVOC emissions were reduced in spring and summer compared to natural
drought whereas, in autumn, an increase was observed for some compounds. These results are in opposition with
the results obtained after only one year of amplified drought (2012), especially for isoprene, where an increase
was observed for this compound (Génard-Zielinski *et al.* in prep). Amplified drought did not seem to shift the
dependence to light and/or temperature which remained unchanged between treatments.
Moreover, two different dependence behaviours were found: (i) all six BVOCs monitored showed daytime light
and temperature dependencies while (ii) only three BVOCs (methanol, acetone and formaldehyde) also showed
that their emissions were maintained during the night with no light at rather constant nocturnal temperatures.
Moreover, some phenomena, such as methanol and acetaldehyde emissions bursts in early morning or the
formaldehyde deposition/uptake (formaldehyde), were not assessed by either $L+T$ or $T$ algorithm.

## Appendix A: calculation of ecophysiological parameters

Net photosynthesis ($Pn$, $\mu molCO_2$ $m^{-2}$ $s^{-1}$) was calculated using equations described by Von Caemmerer and
Farquhar (1981) as follows:

$$\mathrm{Pn} = \frac{F*(Cr-Cs)}{S} - CS*E \tag{A1}$$

Where $F$ is the inlet air flow (mol $s^{-1}$), $Cs$ and $Cr$ are the sample and reference $CO_2$ molar fraction respectively
(ppm), $S$ is the leaf surface ($m^2$), $Cs*E$ is the fraction of $CO_2$ diluted in water evapotranspiration and $E$ (molH$_2$O
$m^{-2}$ $s^{-1}$ then transformed in mmolH$_2$O $m^{-2}$ $s^{-1}$, afterward) is the transpiration rate calculated as follow:

$$\mathrm{E} = \frac{F*(Ws-Wr)}{S*(1-Ws)} \tag{A2}$$

where $Ws$ and $Wr$ are the sample and the reference $H_2O$ molar fraction respectively (molH$_2$O mol$^{-1}$).
Stomatal conductance to water ($Gw$, molH$_2$O $m^{-2}$ $s^{-1}$ then transformed in mmolH$_2$O $m^{-2}$ $s^{-1}$) was calculated using
the following equation:

$$\mathrm{Gw} = \frac{E*(1-\frac{Wl-Ws}{2})}{Wl-Ws} \tag{A3}$$

where $Wl$ is the molar concentration of water vapour within the leaf (molH$_2$O mol$^{-1}$) calculated as follows:

$$Wl = \frac{Vpsat}{P} \tag{A4}$$

where Vpsat is the saturated vapour pressure (kPa) and P was the atmospheric pressure (kPa).

## Appendix B: Modelled emissions calculation

The modelled emissions rates according to light and temperature ($ER_{L+T}$) or the temperature algorithm ($ER_T$) were
calculated according to algorithms described in Guenther *et al.* (1995) as follows :
$$ER_{L+T} = EF_{L+T} * C_l * C_t \tag{B1}$$
where $EF_{L+T}$ is the emission factor at 1000 µmol m$^{-2}$ s$^{-1}$ of photosynthetically active radiation (PAR) and 30°C of
temperature (obtained with the slope of the correlation between experimental emissions and $C_l*C_t$ without forcing
data to pass through the origin), $C_l$ and $C_t$ correspond to light and temperature dependence factors respectively and
were calculated with the following formulae:
$$C_l = \frac{\alpha C_{L1} L}{\sqrt{1 + \alpha^2 L}} \tag{B2}$$
$$C_t = \frac{exp \frac{C_{T1}(T - T_S)}{R T_S T}}{1 + exp \frac{C_{T2}(T - T_M)}{R T_S T}} \tag{B3}$$
where $\alpha = 0.0027$, $C_{L1} = 1.066$, $C_{T1} = 95000$J mol$^{-1}$, $C_{T2} = 230000$J mol$^{-1}$, $T_M = 314$K are empirically derived
constants, $L$ is the photosynthetically active radiation (PAR) flux (µmol m$^{-2}$ s$^{-1}$), $T$ is the leaf experimental
temperature (K) and $T_S$ is the leaf temperature at standard condition (303K).
Modelled emissions according to temperature alone that is $ER_T$, was calculated as follows:
$$ER_T = EF_T * C_T \tag{B4}$$
where $EF_T$ is the emission factor at 30°C of temperature (obtained with the slope of the correlation between
experimental emissions and $C_T$ without forcing data to pass through the origin) and $C_T$ is a temperature dependence
factor calculated as follows:
$$C_T = \exp[\beta(T - T_S)] \tag{B5}$$
where β is an empirical coefficient (with a standard variation value of 0.09K$^{-1}$ used in literature when not measured)
determined, in this study, for each compound according to the season and the treatment through the slope of the
correlation between the natural logarithm of measured emissions rates (ER, µgC g$_{DM}^{-1}$ h$^{-1}$) and experimental
temperature (expressed in K), $T$ is the leaf experimental temperature (K) and $T_S$ is the standard temperature (303K).

**Author contribution**

AS, EO and CF designed the research and the experimental design. AS, BTR, EO and CF conducted the research.
AS, CB, BTR, and CL collected and analyzed the data. AS, EO, CB, HW, BTR, AA and CF wrote the manuscript

**Competing interests**

The authors declare that they have no conflict of interest.

**Acknowledgments**

This work was supported by the French National Agency for Research (ANR) through the SecPriMe² project
(ANR-12-BSV7-0016-01); Europe (FEDER) and ADEME/PACA for PhD funding. We are grateful to FR3098
ECCOREV for the O₃HP facilities (https://o3hp.obs-hp.fr/index.php/fr/). We are very grateful to J.-P. Orts, I.
Reiter. We also thank all members of the DFME team from IMBE and particularly: S. Greff, S. Dupouyet and A.
Bousquet-Melou for their help during measurements and analysis. We thank also, the Université Paris Diderot-

Paris7 for its support. The authors thank the MASSALYA instrumental platform (Aix Marseille Université, lce.univ-amu.fr) for the analysis and measurements used in this publication.

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

**Table:**
**Table 1: N**et photosynthesis ($Pn$, $\mu molCO_2$ $m^{-2}$ $s^{-1}$), stomatal conductance to water ($Gw$, $mmolH_2O$ $m^{-2}$ $s^{-1}$) and emission rates ($\mu gC$ $g_{DM}^{-1}$ $h^{-1}$) according to treatment and season.
Values represent an average of all data measured between 12h and 15h (local time). Letters denote the difference between drought treatments with a > b and values showed
represent the mean ± SE, n=5. ND: natural drought and AD: amplified drought with ns = non-significant, $^{(*)} = 0.05 < P < 0.1$, $* = 0.01 < P < 0.05$, $** = 0.001 < P < 0.01$,

| Season | Spring | | | Summer | | | Autumn | | |
|---|---|---|---|---|---|---|---|---|---|
| Treatments | ND | AD | *P* | ND | AD | *P* | ND | AD | *P* |
| **Pn** | 11 ± 1 a | 9 ± 2 a | ns | 14 ± 2 a | 9±1.2 b | $^{(*)}$ | 7 ± 1 a | 9 ± 1 a | ns |
| **Gw** | 110 ± 19 a | 57 ± 13 b | $^{(*)}$ | 285 ± 38 a | 126 ± 17 b | ** | 122 ± 23 a | 74 ± 21 a | ns |
| **Isoprene** | 20 ± 4 a | 10 ± 2 b | * | 124 ± 10 a | 81 ± 11 b | * | 3 ± 1 a | 5 ± 2 a | ns |
| **MACR+MVK+ISOPOOH** | 0.1 ± 0.03a | 0.1 ± 0.01 a | ns | 0.4 ± 0.1 a | 0.2 ± 0.02 b | * | 0.04 ± 0.01 a | 0.1 ± 0.01 a | ns |
| **Methanol** | 1 ± 0.1 a | 0.5 ±0.04 b | * | 1 ± 0.2 a | 0.6 ± 0.03 b | * | 0.2 ± 0.03 a | 0.2 ± 0.1 a | ns |
| **Acetaldehyde** | 1 ± 0.4 a | 1 ± 0.3 a | ns | 2 ± 0.5 a | 1 ± 0.1 a | ns | 1 ± 0.3 a | 1 ± 0.3 a | ns |
| **Acetone** | 0.5 ± 0.1 a | 0.2 ± 0.02 a | ns | 1 ± 0.2 a | 0.5 ± 0.04 b | ** | 0.4 ± 0.1 a | 0.4 ± 0.1 a | ns |
| **Formaldehyde** | 0.2 ± 0.05 a | 0.1 ± 0.01 a | ns | 0.4 ± 0.1 a | 0.1 ± 0.02 b | ** | 0.2 ± 0.1 a | 0.3 ± 0.1 a | ns |


 **Figure legends**

**Figure 1**: Ombrothermic diagram for natural and amplified drought in 2012, 2013 and 2014. Bars represent mean
monthly precipitation (mm) and curves represent mean monthly temperature (°C). On each amplified drought
graph, the percentage represents the proportion of excluded rain compared to the natural drought plot.

**Figure 2**: Diurnal pattern of stomatal conductance ($Gw$) and net photosynthesis ($Pn$) according to drought
treatment and season**.** Values showed represent means ± SE, n=5.

**Figure 3**: Diurnal pattern of isoprene emissions rates, where points represent measured emission and the yellow
line corresponds to modelled emissions rates according to the $L+T$ algorithm ($ER_{L+T}$). R² and slope (sl) of
correlations between measured (x axis) and modelled (y axis) emissions are presented in the yellow frame.
Correlations were obtained without forcing data to pass through the origin. Values are mean ± SE, n=5.

**Figure 4**: Diurnal pattern of acetaldehyde emissions rates, where points represent measured emission, the yellow
line corresponds to modelled emissions rates according to the $L+T$ algorithm ($ER_{L+T}$) and the dotted line
corresponds to modelled emissions rates according to the $T$ algorithm ($ER_T$). R² and slope (sl) of correlations
between measured (x axis) and modelled (y axis) emissions are presented in the yellow frame for $L+T$ and in the
white frame for $T$. Correlations were obtained without forcing data to pass through the origin. Values are mean ±
SE, n=5.

**Figure 5**: Diurnal pattern of measured methanol emissions rates. Points represent measured emission, the yellow
line corresponds to modelled emissions rates according to the $L+T$ algorithm ($ER_{L+T}$) and the dotted line
corresponds to modelled emissions rates according to the $T$ algorithm ($ER_T$). R² and slope (sl) of correlations
between measured (x axis) and modelled (y axis) emissions are presented in the yellow frame for $L+T$ and in the
white frame for $T$. Correlations were obtained without forcing data to pass through the origin. Values are mean ±
SE, n=5.

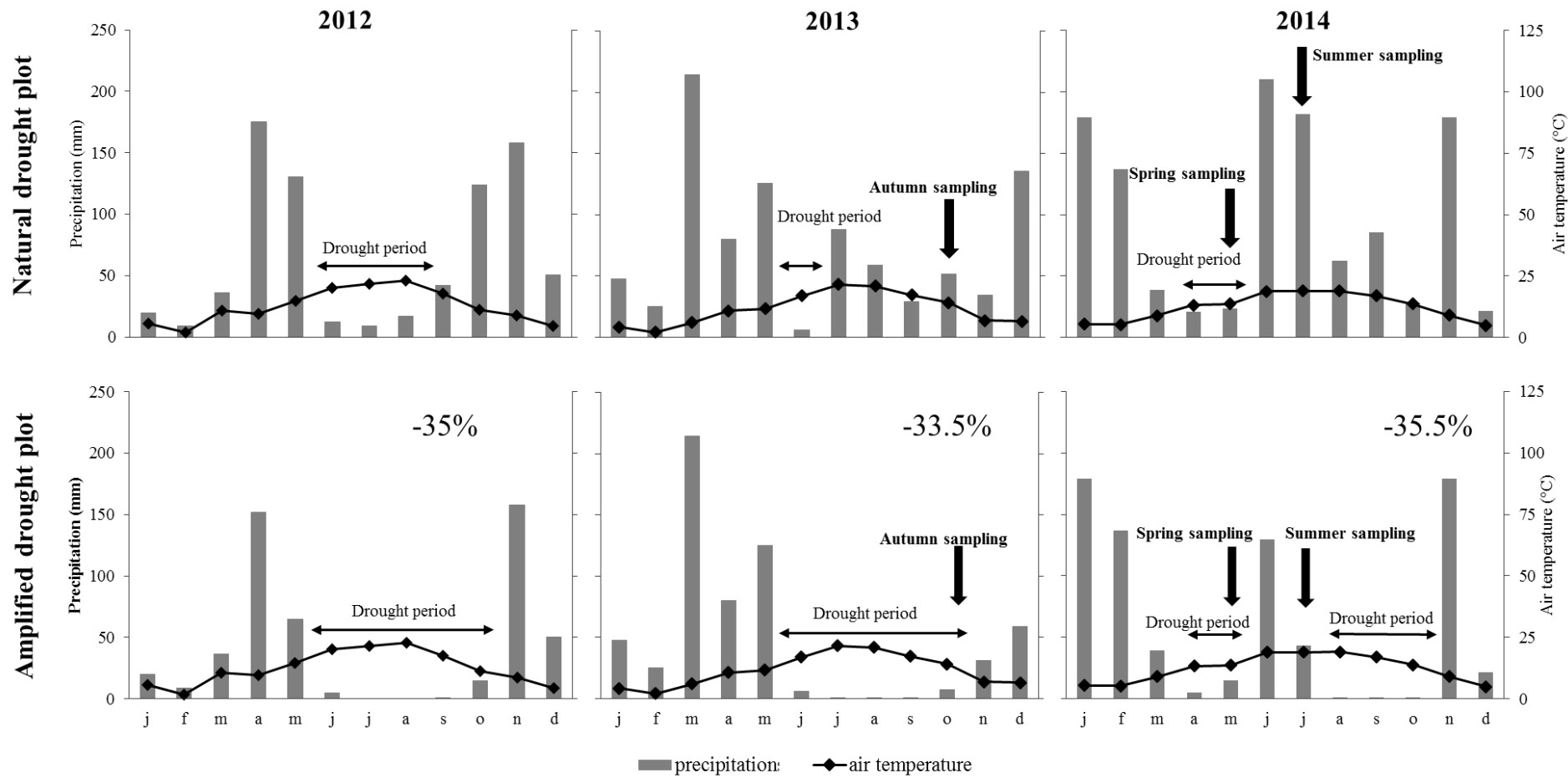


**Figure 1:**




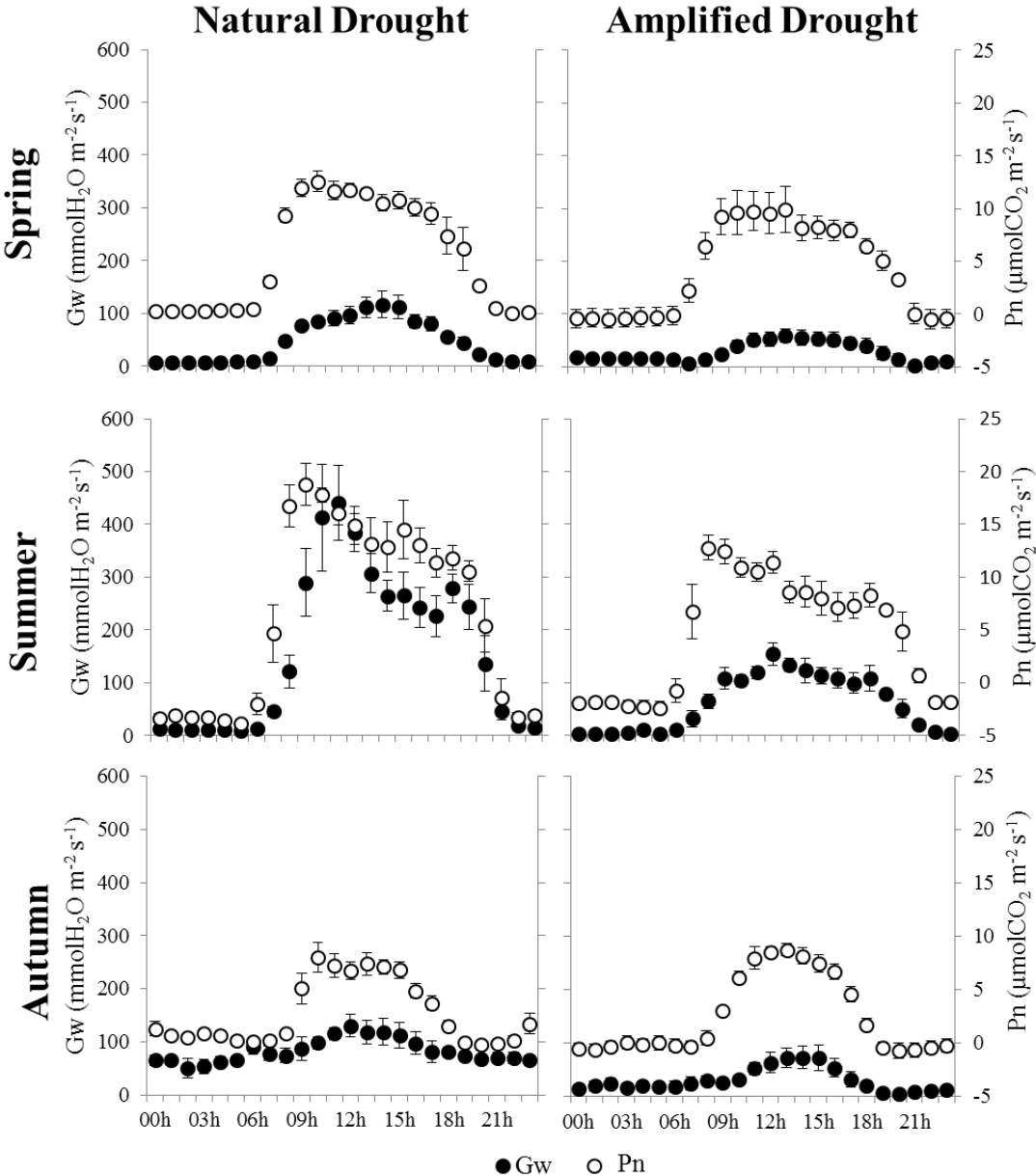


**Figure 2:**

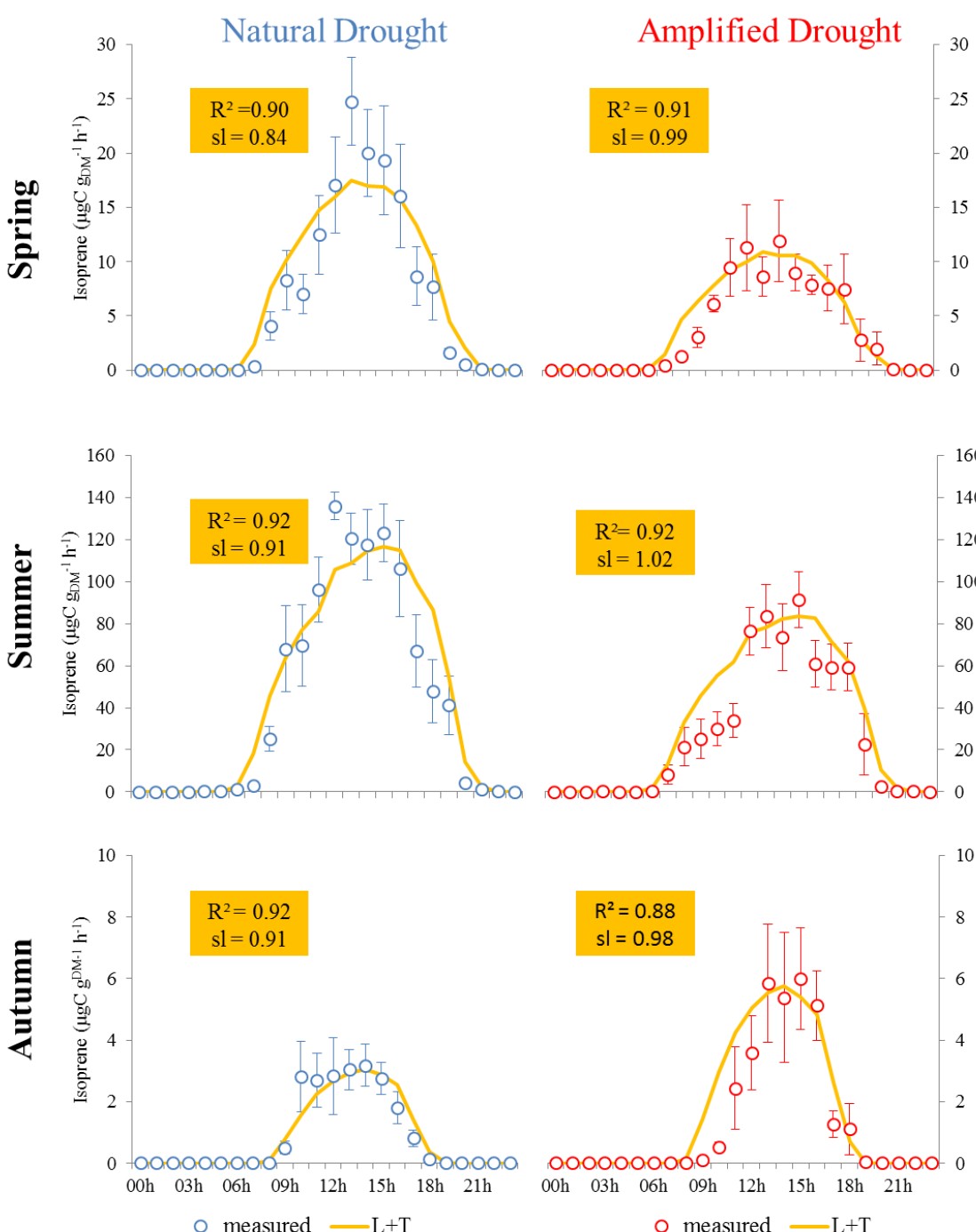


**Figure 3:**


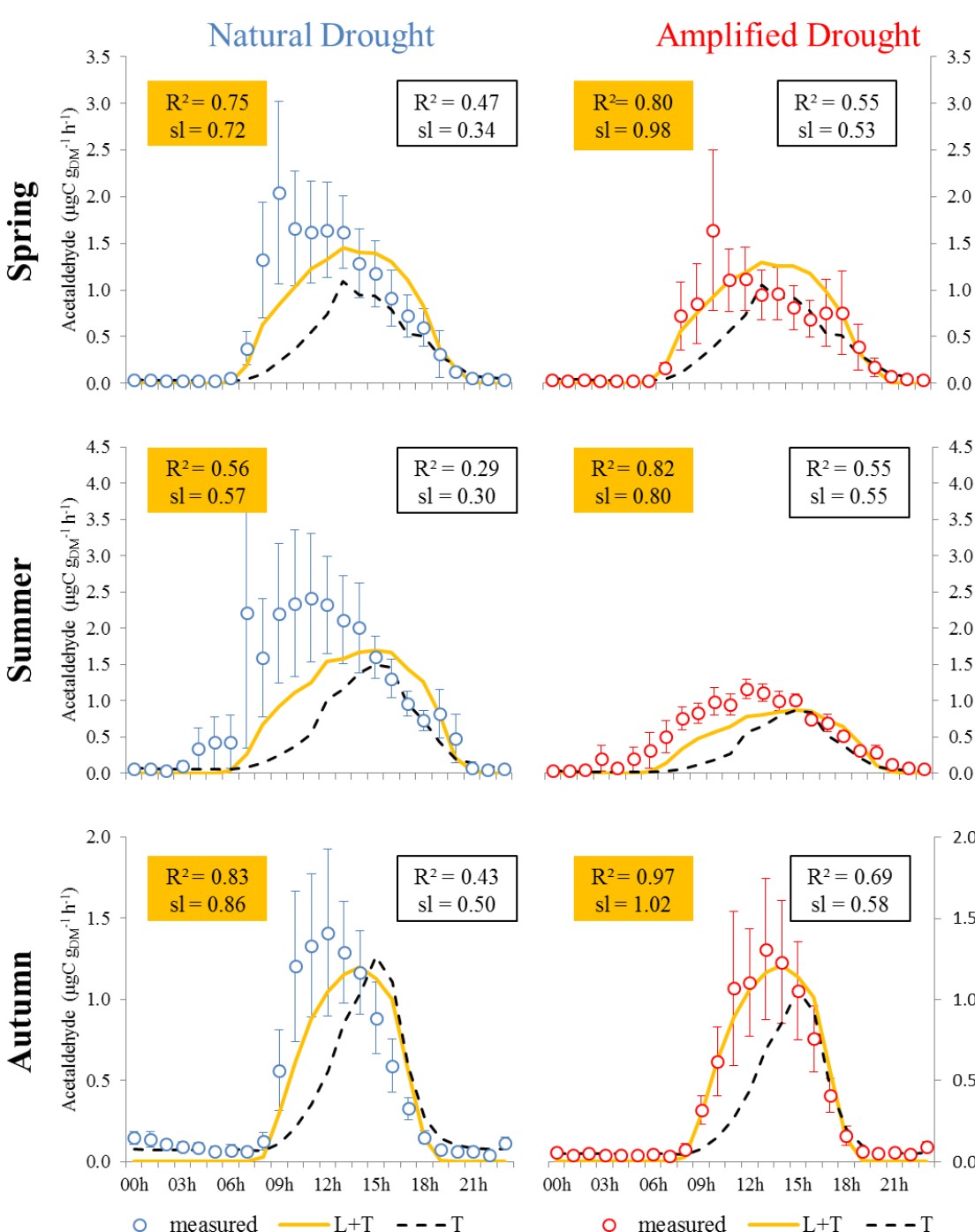


**Figure 4:**

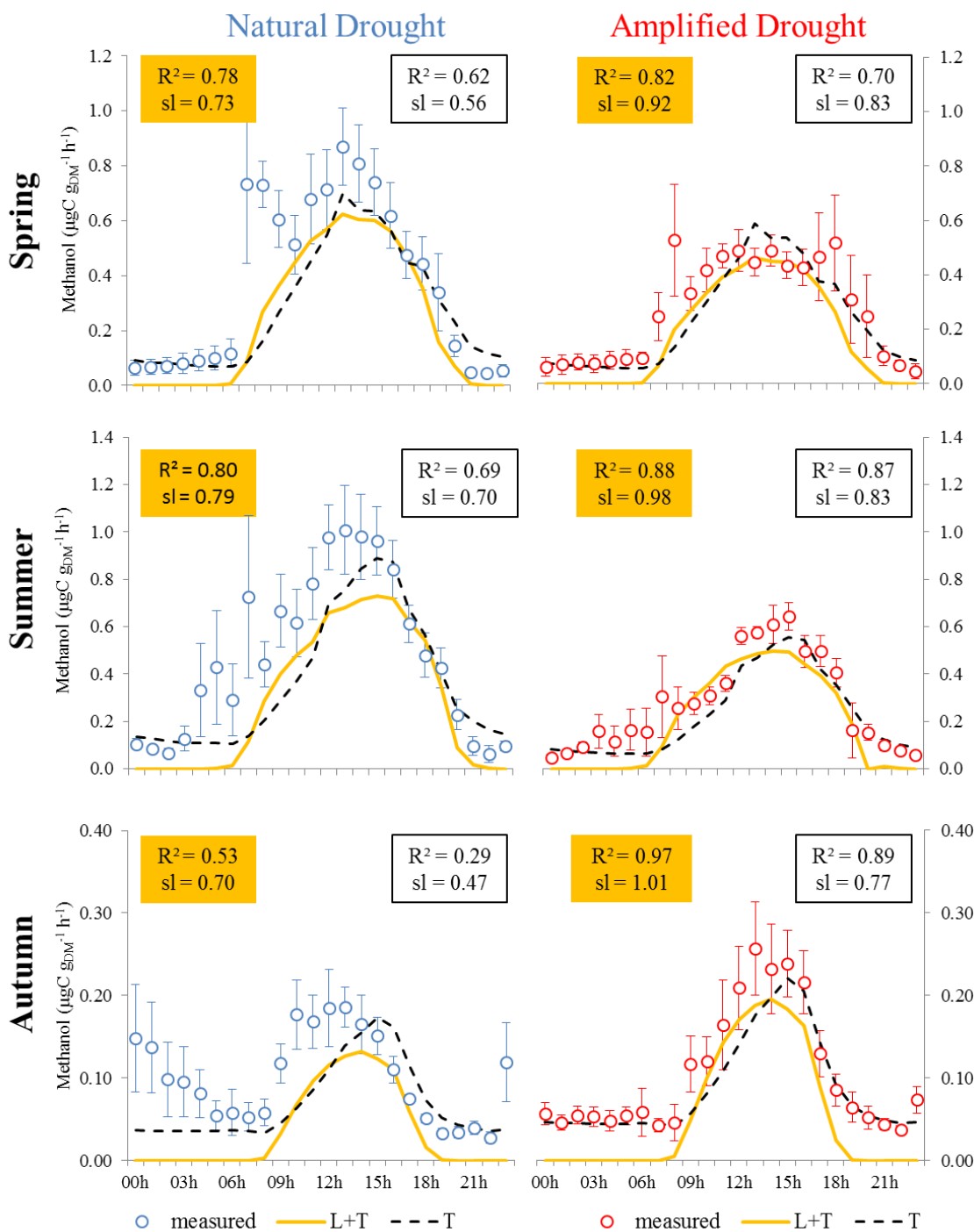


**Figure 5:**



