# Peer review of "Effect of mid-term drought on Quercus pubescens BVOCs"

_Atmospheric Chemistry and Physics, 2016_

## Referee Comment (RC1) · Anonymous Referee #1 · 10 Nov 2016

General comments

The manuscript discusses the light and temperature dependencies of several BVOC emissions from a Mediterranean oak species. This manuscript fits in the scope of the journal presenting a BVOC emission study on a relatively little studied tree species. The authors go through the methods they have used thoroughly, and the results are presented in the text and figures clearly. The discussion on the results and conclusions could, however, be deeper and underline how this study increases the understanding of BVOC emission dynamics. Though the manuscript is carefully written, some English language improvement would not be bad idea. My comments below are rather minor though their number is relatively high.

Specific comments

Line 13: You discuss many times about BVOC in singular form, though you actually mean plural BVOCs. Please check these throughout the text.

Line 23: You claim that the three sampling campaigns cover the entire seasonal cycle. However, note that there are likely sub-seasonal periods, which are not covered by your measurements. For example, the highest natural drought at the site is likely in late summer, when you did not measure. Do you think that your results from these three measurement periods are representative enough to model Q. pubescens BVOC emissions year around? If so, why? Could you describe with a few words the physiological state of the oaks during each of the campaigns, e.g. if the new leaf emergence or leaf size growth occurred during the spring measurement period?

Line 24: Amplified drought impacted all studied BVOCs, but not necessarily all the minor compounds that the trees produce but you couldn't quantify.

Line 32: Please use throughout the text the unit formatting as advised in the journal instructions.

Line 34: Please check the use of subscripts in the entire text.

Line 35: You likely mean tropospheric ozone concentration.

Lines 72-74: In my mind, seven commas per a sentence is too much and makes the sentence hard to read. Please edit the sentence e.g.: However, there are still some misunderstandings at the level of emission mechanisms and consequently on model estimations for isoprene and, a fortiori, for highly volatile BVOCs under mild or severe water stress. In addition, you could open which misunderstandings you mean here.

Line 80-81: Please correct: 2 million ha. Note that the study by Keenan et al. (2009) considers only forests, and there are other remarkable sources as well.

Line 86: The site may be free from direct human disturbance, but indirect disturbance

through e.g. air pollution it certainly has experienced.

Line 93: The plots were 200-300 m2 in size. How many trees were growing in the plots? Can you be sure that the trees at the amplified drought plot did not uptake water by their vast root system from the non-drought area?

Lines 96-97: I do not quite hit the idea of the latter part of the sentence: – corresponding for three years, to 2 months for natural treatment and 5 months for amplified treatment of drought period. Please rephrase.

Line 100: You had five trees per treatment, but how many enclosures there were per tree and per sampling campaign? Did you move enclosures from tree to tree during one sampling campaign?

Line 103: To be precise, BVOC exchange between the tree and the atmosphere is a part of tree gas exchange.

Line 104: How much biomass the enclosures enclosed? Please give some numbers (branch length, leaf area, leaf mass or equivalent).

Line 106: A PTFE air generator sounds like it would produce PTFE in the air. Please rephrase.

Line 109: What do you mean by the excess of air humidity? Was the humidity inside the enclosure controlled (currently not stated in the text) and set to some range? If so, please make an addition in the text, as this is rather critical detail in the case of water-soluble compounds.

Line 116: Rather say: made of PTFE.

Line 119: Is reference to chapter 2.2 correct or should it refer to 2.4 (BVOC analysis)?

Line 120: Please edit: gas exchange values.

Line 125: Add s: parameters. Lyophilization is not familiar term to many readers of the

journal, so say rather: –were lyophilized (freeze-dried) to assess the dry mass.

Line 140-141: You say that formaldehyde calculation took into account the humidity-dependence. What about the other humidity-dependent compounds? Could the clearly visible steps methanol and acetone fluxes in the late evenings of natural drought (fig. 4 and S3) be humidity-related? Anyhow, there seems to be something else happening simultaneously: net photosynthesis rises to positive values just before midnight (fig. 1, autumn, natural drought). Something wrong with the measurements or calculation?

Line 145: Why did you choose to express the emission rates as C (carbon)?

Line 164-165: Please rephrase for example as follows: Afterwards, linear regression tests and slope tests (equal to 1) were also performed.

Line 168: Have you any data how dry the soil actually was? Any soil volumetric water content measurements or equivalent throughout the seasons?

Line 171: Please correct spelling: other season and stomatal closure (the latter one in some following lines as well).

Line 177: I wonder if you have any tree growth data from the site? In ceasing growth (height growth or lateral growth depending on timing) you might see drought effect earlier than in photosynthesis. The results are not discussed and compared to literature too much, so you could here e.g. refer to an earlier drought study (Damesin & Rambal 1995) conducted with the same species.

Line 186-187: Reduced and increased emissions compared to what? And what is the reference for? In the discussion about isoprene emission dynamics during drought, you may also refer to Bruggemann & Schnitzler (2002), who have studied isoprene emissions of Q. pubescens saplings.

Line 193: You write here and in many other cases as well, that a compound responds to something. This reflects very much the modelling point of view. However, the plant responds to the changes in its environment, and that we see as a change in the plant

volatile emissions. I would like to see in the discussion more of this plant-point-of-view: what does the plant do so that we see these kind of fluxes.

Line 196-199: You write: "the daily cycle between natural and amplified drought was very different for each season." If I look at the fig. 2 about isoprene emissions, I don't see very different daily cycles. Please clarify what you mean. Moreover, you write: "were not the only parameters driving isoprene emissions." Please tell which other parameters you think were affecting at that time of the year.

Line 200: You discuss about MACR+MVK+ISOPOOH basically as a compound. Have you any data if all these three compounds really are present in the fluxes all the time or if one of them dominates the measured flux and thus masks the variations in the others?

Line 213: Turn the sign: <.

Line 221: Please check spelling: phenomena.

Line 227: Please change to leaf elongation.

Line 230: You write that methanol emissions respond only to temperature in nighttime. Have you taken into account that in nighttime light intensity is basically zero if no artificial light is available and stays constant over the night? Moreover, in nighttime light intensity range is far smaller than in daytime, and this will be reflected in your modelling results.

Line 254-255: Would this sentence need a reference?

Line 261: Please change phenomenon to phenomena.

Line 263: Please check spelling: the calculation of ecophysiological parameters.

Line 278: Please check spelling: vapour.

Line 327-328: Here and in some other cases as well the italics of scientific names have

been replaced with cryptic markings. Please check the reference list.

Table 1 caption: Please remove the abbreviation ER and add the explanations for ND and AD.

Figure 1: Please remove "ND: natural drought; AD: aggravated drought" as the information is in the figure. The various vertical scales make it hard to compare the seasons, so please consider unifying the scales. And please remove A from the lower right panel.

Figure 2-4 captions: Edit the last sentences: – emissions are presented –.

References

Bruggemann, N., Schnitzler, J.P., 2002. Comparison of isoprene emission, intercellular isoprene concentration and photosynthetic performance in water-limited oak (Quercus pubescens Willd. and Quercus robur L.) saplings. Plant Biol. 4, 456-463. http://dx.doi.org/10.1055/s-2002-34128.

Damesin, C., Rambal, S., 1995. Field study of leaf photosynthetic performance by a Mediterranean deciduous oak tree (Quercus pubescens) during a severe summer drought. New Phytol. 131, 159-167. http://dx.doi.org/10.1111/j.1469-8137.1995.tb05717.x.
* * *

---

## Author Comment (AC1) · 12 Dec 2016

The authors would like to thank the Editor and the Reviewers for very careful and detailed review of our manuscript and providing of comments and suggestions to improve the quality of the manuscript.

RC1: The manuscript discusses the light and temperature dependencies of several BVOC emissions from a Mediterranean oak species. This manuscript fits in the scope of the journal presenting a BVOC emission study on a relatively little studied tree species. The authors go through the methods they have used thoroughly, and the results are presented in the text and figures clearly. The discussion on the results and conclusions could, however, be deeper and underline how this study increases the un-

derstanding of BVOC emission dynamics. Though the manuscript is carefully written, some English language improvement would not be bad idea. My comments below are rather minor though their number is relatively high.

RC1: Line 13: You discuss many times about BVOC in singular form, though you actually mean plural BVOCs. Please check these throughout the text. AC: We did the modification in the manuscript.

RC1: Line 23: You claim that the three sampling campaigns cover the entire seasonal cycle. However, note that there are likely sub-seasonal periods, which are not covered by your measurements. For example, the highest natural drought at the site is likely in late summer, when you did not measure. Do you think that your results from these three measurement periods are representative enough to model Q. pubescens BVOC emissions year around? If so, why? Could you describe with a few words the physiological state of the oaks during each of the campaigns, e.g. if the new leaf emergence or leaf size growth occurred during the spring measurement period? AC: We think that our measurement periods are well representative of Q. pubescens BVOCs emissions because they took place during the principal phenological stages of leaves. There was no leaf emergence during our spring period sampling, it was the end of leaf growth. We also performed an experiment during leaf emergence (in April 2013), not presented in this study, but there were very slight BVOCs emissions. During summer, leaves still matured and autumn was featured by the beginning of leaf senescence. We added a short comment on this part in the manuscript (lines 105-106).

RC1: Line 24: Amplified drought impacted all studied BVOCs, but not necessarily all the minor compounds that the trees produce but you couldn't quantify. AC: Indeed, we added "studied" in this line.

RC1: Line 32: Please use throughout the text the unit formatting as advised in the journal instructions. AC: We did the modification in the manuscript.

RC1: Line 34: Please check the use of subscripts in the entire text. AC: We did the

modification in the manuscript.

RC1: Line 35: You likely mean tropospheric ozone concentration. AC: Yes, we changed this in the manuscript (line 36).

RC1: Lines 72-74: In my mind, seven commas per a sentence is too much and makes the sentence hard to read. Please edit the sentence e.g.: However, there are still some misunderstandings at the level of emission mechanisms and consequently on model estimations for isoprene and, a fortiori, for highly volatile BVOCs under mild or severe water stress. In addition, you could open which misunderstandings you mean here. AC: We rewrote this sentence as suggested and we added some details on misunderstandings on isoprene emissions (lines 66-74).

RC1: Line 80-81: Please correct: 2 million ha. Note that the study by Keenan et al. (2009) considers only forests, and there are other remarkable sources as well. AC: We did the modification in the manuscript (line 84).

RC1: Line 86: The site may be free from direct human disturbance, but indirect disturbance through e.g. air pollution it certainly has experienced. AC: Indeed, we added "direct" in this sentence (line 90).

RC1: Line 93: The plots were 200-300 m2 in size. How many trees were growing in the plots? Can you be sure that the trees at the amplified drought plot did not uptake water by their vast root system from the non-drought area? AC: In natural drought plot, there is 145 stems and in amplified drought plot, there is 95 stems. We cannot be sure that trees at the amplified drought did not uptake water in natural drought plot. Indeed, we do not know where the trees roots are located. But, on our site, there is a buffer zone for each plot (2 meters). So, we only followed trees located in the heart of both plots. And, also, we observed effect of amplified drought on BVOC emissions and physiology parameters. We think that it a negligible phenomenon.

RC1: Lines 96-97: I do not quite hit the idea of the latter part of the sentence: – corresponding for three years, to 2 months for natural treatment and 5 months for amplified treatment of drought period. Please rephrase. AC: We rephrased this sentence (lines 101-102).

RC1: Line 100: You had five trees per treatment, but how many enclosures there were per tree and per sampling campaign? Did you move enclosures from tree to tree during one sampling campaign? AC: During each field campaigns, the five trees of each plot were sampled. We used 2 enclosure systems concomitantly which allowed us to analyze one tree under amplified drought and one tree under natural drought. The analyses was performed during 1 or 2 days, depending on the weather. And, every 1 or 2 days, we moved enclosures from tree to tree. We added a sentence about that in the manuscript (lines 111-112).

RC1: Line 103: To be precise, BVOC exchange between the tree and the atmosphere is a part of tree gas exchange. AC: We did the modification in the manuscript.

RC1: Line 104: How much biomass the enclosures enclosed? Please give some numbers (branch length, leaf area, leaf mass or equivalent). AC: We enclosed branches containing between 8 and 12 leaves which corresponded to 1.4g and 3.6g of dry matter. In terms of surface, we enclosed between 110 and 320 $cm^2$ of leaves. With these data, we calculated the relation between leaf mass and surface (LMA) and we found no significant difference between leaves from amplified and natural drought at any season. We added a sentence about that (lines 125-127).

RC1: Line 106: A PTFE air generator sounds like it would produce PTFE in the air. Please rephrase. AC: We rephrased this sentence (line 113).

RC1: Line 109: What do you mean by the excess of air humidity? Was the humidity inside the enclosure controlled (currently not stated in the text) and set to some range? If so, please make an addition in the text, as this is rather critical detail in the case of water-soluble compounds. AC: The humidity inside the enclosure was not controlled. However, we slightly removed some humidity from entering the chambers

(before the air generator), especially in autumn to impede condensation of water vapor which would have disturb mass flow controller.

RC1: Line 116: Rather say: made of PTFE. AC: We did the modification in the manuscript (line 113).

RC1: Line 119: Is reference to chapter 2.2 correct or should it refer to 2.4 (BVOC analysis)? AC: Indeed, we did the modification in the manuscript (line 130).

RC1: Line 120: Please edit: gas exchange values. AC: We did the modification in the manuscript (line 131).

RC1: Line 125: Add s: parameters. Lyophilization is not familiar term to many readers of the journal, so say rather: –were lyophilized (freeze-dried) to assess the dry mass. AC: We did the modification in the manuscript (line 136).

RC1: Line 140-141: You say that formaldehyde calculation took into account the humidity dependence. What about the other humidity-dependent compounds? Could the clearly visible steps methanol and acetone fluxes in the late evenings of natural drought (fig.4 and S3) be humidity-related? Anyhow, there seems to be something else happening simultaneously: net photosynthesis rises to positive values just before midnight (fig. 1, autumn, natural drought). Something wrong with the measurements or calculation? AC: We only took into account the humidity dependency of formaldehyde because for this compound, this dependency was very problematic compared to the others compounds (Vlasenko et al. 2010). We do not think that the increase of methanol and acetone in late evening (in autumn) could come from the humidity because we analyzed a pair of trees at a time (one tree under natural drought and one tree under amplified drought). Moreover, the enclosure chambers were feed with the same inlet air (thus, with a similar humidity) and transpiration rate during the night was close to zero. If there was a humidity problem, with our set-up, we would have observed the same phenomenon on amplified drought and it was not the case. Moreover, it seems unlikely that there was a calculation problem because we always used

the same calculation. It was probably a phenomenon linked to trees metabolisms but we cannot explain this yet.

RC1: Line 145: Why did you choose to express the emission rates as C (carbon)? AC: We chose to express the emission rates in carbon because in many studies of dealing with BVOC modelling, they used this unit (Guenther et al. 2012; Guenther 2013). Also, in global scale, it is more convenient to express BVOC emissions in carbon to evaluate their impact on global change.

RC1: Line 164-165: Please rephrase for example as follows: Afterwards, linear regression tests and slope tests (equal to 1) were also performed. AC: We did the modification in the manuscript (lines 178-179).

RC1: Line 168: Have you any data how dry the soil actually was? Any soil volumetric water content measurements or equivalent throughout the seasons? AC: We have predawn water potential only for the summer campaign which can give a good idea of water availability in soil. During this season, there is a significant difference between both plots (-0.61MPa for natural drought and -0.85MPa for amplified drought, P < 0.05). Moreover, we observed an effect of our treatments in physiology, especially on stomatal conductance.

RC1: Line 171: Please correct spelling: other season and stomatal closure (the latter one in some following lines as well). AC: We did the modification in the manuscript.

RC1: Line 177: I wonder if you have any tree growth data from the site ? In ceasing growth (height growth or lateral growth depending on timing) you might see drought effect earlier than in photosynthesis. The results are not discussed and compared to literature too much, so you could here e.g. refer to an earlier drought study (Damesin & Rambal1995) conducted with the same species. AC: Indeed, we have some data on tree growth (in terms of leaf biomass and lateral growth) but with no change in 2013 and 2014 and significant reduction of growth in 2016 (data showed on other publication) that is the fifth year of amplified drought. Photosynthesis is typically, is the first parameter

to be impacted by drought (Chaves et al. 2002). That is exactly what occurred in our study because we observed reduction of photosynthesis until 2012 (the first year of our experiment) whereas the first effect on growth appeared in 2016.

RC1: Line 186-187: Reduced and increased emissions compared to what? And what is the reference for? In the discussion about isoprene emission dynamics during drought, you may also refer to Bruggemann & Schnitzler (2002), who have studied isoprene emissions of Q. pubescens saplings. AC: This experiment was conducted since 2012. In this paper, we only presented the results from the end of the second year to the beginning of the third year. In the first year, an increase of isoprene emissions was observed (data unpublished yet) whereas, we observed a decrease after 2-3 years of amplified drought. We added also a sentence on Bruggermann and Schnitzler's work (line 202).

RC1: Line 193: You write here and in many other cases as well, that a compound responds to something. This reflects very much the modelling point of view. However, the plant responds to the changes in its environment, and that we see as a change in the plant volatile emissions. I would like to see in the discussion more of this plant-point-of-view: what does the plant do so that we see these kind of fluxes. AC: We added some part on plant-point of view throughout the discussion.

RC1: Line 196-199: You write: "the daily cycle between natural and amplified drought was very different for each season." If I look at the fig. 2 about isoprene emissions, I don't see very different daily cycles. Please clarify what you mean. Moreover, you write: "were not the only parameters driving isoprene emissions." Please tell which other parameters you think were affecting at that time of the year. AC: Accordingly to the reviewer's comment, we change this sentence since indeed our description was confused. We should have written the daily cycle between natural and amplified drought was different. What was different but the intensity of isoprene emissions between amplified and natural drought. We suggest that plant likely needed to produce more isoprene with the aim to protect the photosystems apparatus in new leaves. We

added this point in the manuscript (lines 217-218).

RC1: Line 200: You discuss about MACR+MVK+ISOPOOH basically as a compound. Have you any data if all these three compounds really are present in the fluxes all the time or if one of them dominates the measured flux and thus masks the variations in the others? AC: We did not have data on these compounds separately. We only detected the ion 73 corresponding on the three compounds. Thus, we cannot say if one of these compounds dominated flux.

RC1: Line 213: Turn the sign: <. AC: In this line, it is the good sign. It was just for specifying that the slope was not significantly different to 1. Maybe, it is confusing and we can remove this indication.

RC1: Line 221: Please check spelling: phenomena. AC: We did the modification in the manuscript (line 239).

RC1: Line 227: Please change to leaf elongation. AC: We did the modification in the manuscript (line 248).

RC1: Line 230: You write that methanol emissions respond only to temperature in nighttime. Have you taken into account that in nighttime light intensity is basically zero if no artificial light is available and stays constant over the night? Moreover, in nighttime light intensity range is far smaller than in daytime, and this will be reflected in your modelling results. AC: We measured light during the night and used these data for modelling. The data on light during nighttime was close to zero and temperature was roughly constant. Thus, we attributed emissions of methanol during the nighttime to a temperature-driven response as already demonstrated by Smiatek and Steinbrecher (2006). We made some figures in the new version of the supplementary files, summarizing light and temperature conditions during our experiment.

RC1: Line 254-255: Would this sentence need a reference? AC: These results were not published yet. Thus, we added personal communication from A.C. Génard-

Zielinski, line 282).

RC1: Line 261: Please change phenomenon to phenomena. AC: We did the modification in the manuscript (line 289).

RC1: Line 263: Please check spelling: the calculation of ecophysiological parameters. AC: We did the modification in the manuscript (line 291).

RC1: Line 278: Please check spelling: vapour. AC: We did the modification in the manuscript (305).

RC1: Line 327-328: Here and in some other cases as well the italics of scientific names have been replaced with cryptic markings. Please check the reference list. AC: We did the modification in the manuscript.

RC1: Table 1 caption: Please remove the abbreviation ER and add the explanations for ND and AD. AC: We did the modification in the manuscript.

RC1: Figure 1: Please remove "ND: natural drought; AD: aggravated drought" as the information is in the figure. The various vertical scales make it hard to compare the seasons, so please consider unifying the scales. And please remove A from the lower right panel. AC: We did the modification in the manuscript.

RC1: Figure 2-4 captions: Edit the last sentences: – emissions are presented –. AC: We did the modification in the manuscript.

Chaves M.M., Pereira J.S., Maroco J., Rodrigues M.L., Ricardo C.P.P., Osório M.L., Carvalho I., Faria T. & Pinheiro C. (2002). How plants cope with water stress in the field? Photosynthesis and growth. Annals of botany, 89, 907-916.

Guenther A. (2013). Biological and Chemical Diversity of Biogenic Volatile Organic Emissions into the Atmosphere. International Scholarly Research Notices, 2013.

Guenther A., Jiang X., Heald C., Sakulyanontvittaya T., Duhl T., Emmons L. & Wang X. (2012). The Model of Emissions of Gases and Aerosols from Nature version 2.1

(MEGAN2. 1): an extended and updated framework for modeling biogenic emissions.

Smiatek G. & Steinbrecher R. (2006). Temporal and spatial variation of forest VOC emissions in Germany in the decade 1994–2003. Atmospheric Environment, 40, 166-177.

Vlasenko A., Macdonald A., Sjostedt S. & Abbatt J. (2010). Formaldehyde measurements by Proton transfer reaction–Mass Spectrometry (PTR-MS): correction for humidity effects. Atmospheric Measurement Techniques, 3, 1055-1062.

Please also note the supplement to this comment:
http://www.atmos-chem-phys-discuss.net/acp-2016-836/acp-2016-836-AC1-supplement.pdf

**Supplement:**

**Table S1**: Values of the paramaters used to calculate the modelled emissions with L+T or T algorithms, standardised emissions factors for $L+T$ algorithm ($EF_{L+T}$), specific standardised emissions factors for $T$ algorithm ($EF_T$) and experimental coefficient β. Means ± se, n = 5.

**Table S1:**

| Compounds | Treatment | Spring $EF_{L+T}$ | $EF_T$ | β | Summer $EF_{L+T}$ | $EF_T$ | β | Autumn $EF_{L+T}$ | $EF_T$ | β |
|---|---|---|---|---|---|---|---|---|---|---|
| **Isoprene** | ND | 28.5 ± 4.6 | | | 118.0 ± 8.4 | | | 6.4 ± 1.1 | | |
| | AD | 17.8 ± 2.4 | | | 84.8 ± 9.0 | | | 12.0 ± 2.8 | | |
| **MACR+MVK+ISOPOOH** | ND | 0.2 ± 0.03 | 0.7 ± 0.15 | 0.5 ± 0.01 | 0.3 ± 0.02 | 0.4 ± 0.1 | 0.5 ± 0.1 | 0.1 ± 0.01 | 1.6 ± 0.3 | 0.7 ± 0.1 |
| | AD | 0.1 ± 0.01 | 0.9 ± 0.28 | 0.6 ± 0.04 | 0.2 ± 0.03 | 0.2 ± 0.04 | 0.6 ± 0.1 | 0.1 ± 0.02 | 5.0 ± 1.6 | 0.9 ± 0.03 |
| **Methanol** | ND | 1.0 ± 0.2 | 2.6 ± 0.8 | 0.3 ± 0.1 | 0.7 ± 0.04 | 0.9 ± 0.04 | 0.3 ± 0.04 | 0.3 ± 0.1 | 1.1 ± 0.3 | 0.4 ± 0.1 |
| | AD | 0.8 ± 0.1 | 2.2 ± 0.2 | 0.3 ± 0.04 | 0.5 ± 0.1 | 0.6 ± 0.04 | 0.3 ± 0.1 | 0.4 ± 0.1 | 1.4 ± 0.4 | 0.4 ± 0.04 |
| **Acetone** | ND | 0.6 ± 0.2 | 1.8 ± 0.5 | 0.4 ± 0.01 | 0.9 ± 0.1 | 1.1 ± 0.2 | 0.4 ± 0.1 | 0.6 ± 0.2 | 2.4 ± 0.6 | 0.4 ± 0.03 |
| | AD | 0.5 ± 0.1 | 2.1 ± 0.4 | 0.5 ± 0.02 | 0.4 ± 0.03 | 0.5 ± 0.1 | 0.3 ± 0.1 | 0.8 ± 0.3 | 4.3 ± 1.8 | 0.5 ± 0.1 |
| **Formaldehyde** | ND | 0.3 ± 0.1 | 0.8 ± 0.2 | 0.4 ± 0.02 | 0.3 ± 0.03 | 0.4 ± 0.1 | 0.4 ± 0.1 | 0.4 ± 0.1 | 1.6 ± 0.4 | 0.4 ± 0.04 |
| | AD | 0.2 ± 0.02 | 1.3 ± 0.2 | 0.5 ± 0.03 | 0.2 ± 0.03 | 0.2 ± 0.1 | 0.4 ± 0.2 | 0.6 ± 0.1 | 2.7 ± 0.7 | 0.5 ± 0.02 |
| **Acetaldehyde** | ND | 2.4 ± 0.7 | 9.4 ± 2.7 | 0.5 ± 0.03 | 1.7 ± 0.4 | 1.6 ± 0.3 | 0.4 ± 0.1 | 2.5 ± 0.6 | 34.0 ± 3.1 | 0.7 ± 0.02 |
| | AD | 2.1 ± 0.9 | 7.9 ± 4.2 | 0.5 ± 0.1 | 0.9 ± 0.1 | 0.9 ± 0.1 | 0.5 ± 0.1 | 2.5 ± 0.7 | 37.8 ± 4.3 | 0.7 ± 0.1 |

**Figure S1**: Diurnal pattern of photosynthetic active radiations (PAR) and temperatures in spring, summer and autumn. Values are mean ± SE, n=5

**Figure S2**: Diurnal pattern of MACR+MVK+ISOPOOH emissions rates, where points represent measured emissions, the yellow line correspond to modelled emissions rates according to the $L+T$ algorithm ($ER_{L+T}$) and dotted line is modelled emissions rates according to $T$ algorithm ($ER_T$). Values are mean ± SE, n=5. $R^2$ and slope (sl) of correlations between measured and modelled emissions were presented in the yellow frame for $L+T$ and in the white frame for $T$. Correlations were obtained without forcing data through the origin.

**Figure S3**: Diurnal pattern of acetone emissions rates where points represent measured emissions, yellow line correspond to modelled emissions rates according to the $L+T$ algorithm ($ER_{L+T}$) and dotted line is modelled emissions rates according to $T$ algorithm ($ER_T$). Values are mean ± SE, n=5. $R^2$ and slope (sl) of correlations between measured and modelled emissions were presented in the yellow frame for $L+T$ and in the white frame for $T$. Correlations were obtained without forcing data through the origin.

**Figure S4**: Diurnal pattern of formaldehyde emissions rates where points represent measured emissions, yellow line correspond to modelled emissions rates according to the $L+T$ algorithm ($ER_{L+T}$) and dotted line is modelled emissions rates according to $T$ algorithm ($ER_T$). Values are mean ± SE, n=5. $R^2$ and slope (sl) of correlations between measured and modelled emissions were presented in yellow frame for $L+T$ and in white frame for $T$. Correlations were obtained without forcing data through the origin.

[Figure]

**Figure S1:**

[Figure]

**Figure S2**:

[Figure]

**Figure S3**:

[Figure]

**Figure S4**:

---

## Referee Comment (RC2) · Anonymous Referee #2 · 17 Dec 2016

This manuscript presents BVOC emission data from the drought tolerant Quercus pubescens using PTR-TOF-MS techniques. The authors study a suite of BVOCs (isoprene, methanol, acetone, acetaldehyde, formaldehyde and MACR+MVK+ISOPOOH) at 3 points over a year, under both natural and amplified drought conditions. They compare observations with model algorithms and report 2 types of emission responses: 1) light and temperature dependent and 2) l/t dependent during the day and only temperature dependent at night.

General Comments English grammar problems are numerous throughout the manuscript.

Your two types of responses can be more easily summarized throughout the

manuscript, "All six BVOCs monitored showed daytime light and temperature dependencies, while three BVOCs (methanol, acetone and formaldehyde) showed nighttime temperature dependencies as well."

Figures 4 and S3 show that the models do accurately simulate the emission burst for methanol as well as the formaldehyde deposition, albeit the models both show a slight lag in the hour of the day in just the autumn natural drought conditions.

Specific comments L13 and throughout manuscript: use plural form "BVOCs" when speaking about more than one compound L19: "...especially in the Mediterranean..." L22: "...a drought tolerant..." L51 – 53: You write: "Several models, already existing (Guenther et al. 2006; Guenther et al. 2012; Menut et al. 2014), predict BVOC emissions according to the type of vegetation, biomass density, leaf age, specific emission factor for many vegetal species, as well as the impact of environmental factors." Please separate references for accuracy. For example, MEAGAN models (Guenther 2006 and 2012 references) do not include vegetation species specific emission factors nor account for leaf age or biomass density.

L70: "...IPCC predicts..." L85: "60 km North of Marseille, France..." L93: You write, "...drought ($300m^2$) and an amplified drought ($232m^2$)." Better indicate what the values in parentheses represent.

L93 – 101: This wording was difficult to understand. How did you determine the extent of drought? How do you know this was indeed a drought stress?

L95 – 97: "During the first year of experiments (2012), 35 % of natural rain was excluded and, afterward, 33.5 and 35.5 % were excluded (2013 and 2014, respectively) corresponding for three years, to 2 months for natural treatment and 5 months for amplified treatment of drought period." This text should be rewritten to clearly describe the differences between the natural and amplified drought treatments in terms of rainfall exclusion and periods of application, i.e. what 2 month period? What 5 month period? Was there any sampling conducted prior to the experiment or during the experiment on

non-drought stressed trees for comparison?

L229 - 231: Nevertheless, our results suggested that methanol emissions responded strongly to light and temperature during the day whereas, during the night, they responded to temperature. See General Comments for suggested clarification.

L261 – 262: "Moreover, some phenomenon, such as the burst in early morning (methanol and acetaldehyde) or the deposition/uptake (formaldehyde), were not modelled by L+T or T algorithm." Figures 4 and S3 show that the models do accurately simulate the emission burst for methanol as well as the formaldehyde deposition, albeit the models both show a slight lag in the hour of the day in just the autumn natural drought conditions

––––––––––––––––––––––––––––––

---

## Author Comment (AC2) · 19 Dec 2016

The authors would like to thank the Editor and the Reviewers for very careful and detailed review of our manuscript and providing of comments and suggestions to improve the quality of the manuscript.

RC2: This manuscript presents BVOC emission data from the drought tolerant Quercus pubescens using PTR-TOF-MS techniques. The authors study a suite of BVOCs (isoprene, methanol, acetone, acetaldehyde, formaldehyde and MACR+MVK+ISOPOOH) at 3 points over a year, under both natural and amplified drought conditions. They compare observations with model algorithms and report 2 types of emission responses: 1) light and temperature dependent and 2) it dependent during the day and only temperature dependent at night.

RC2: General Comments English grammar problems are numerous throughout the manuscript. AC: The manuscript was corrected by a bilingual person.

RC2: Your two types of responses can be more easily summarized throughout the manuscript, "All six BVOCs monitored showed daytime light and temperature dependencies, while three BVOCs (methanol, acetone and formaldehyde) showed nighttime temperature dependencies as well." Figures 4 and S3 show that the models do accurately simulate the emission burst for methanol as well as the formaldehyde deposition, albeit the models both show a slight lag in the hour of the day in just the autumn natural drought conditions. AC: The burst of methanol is observed only in spring and summer between 6am and 8am. For exemple methanol emission reach almost 0.8$\mu$gC.gDM-1.h-1 during the burst contrasting with the previous emission (less than 0.01). The graph shows that none of the model (yellow and dotted lines) fit this burst. Concerning formaldehyde deposition (S4), even if the models follow the same emission pattern than the observations, they don't show any negative emission and so can't accurately allow to estimate the deposition.

RC2: Specific comments L13 and throughout manuscript: use plural form "BVOCs" when speaking about more than one compound. AC: We did the modifications throughout the manuscript.

RC2: L19: ". . .especially in the Mediterranean. . ." AC: We did the modifications throughout the manuscript (line 18).

RC2: L22: ". . .a drought tolerant. . ." AC: We already deleted this point as suggested by the other reviewer.

RC2: L51 – 53: You write: "Several models, already existing (Guenther et al. 2006; Guenther et al. 2012; Menut et al. 2014), predict BVOC emissions according to the type of vegetation, biomass density, leaf age, specific emission factor for many vegetal

species, as well as the impact of environmental factors." Please separate references for accuracy. For example, MEAGAN models (Guenther 2006 and 2012 references) do not include vegetation species specific emission factors nor account for leaf age or biomass density. AC: We did the modification (line 52).

RC2: L70: ". . .IPCC predicts. . ." AC: We did the modification (line 76).

RC2: L85: "60 km North of Marseille, France. . ." AC: We did the modification (line 89).

RC2: L93: You write, ". . .drought (300m2 ) and an amplified drought (232m2 )." Better indicate what the values in parentheses represent. AC: We have corrected this point as follows: "A rainfall exclusion device (an automated monitored roof deployed during selected rain events) was set up over part of the O3HP canopy (232m$^2$ surface) to exclude 30% of raining according to the worst scenario of climate change (Giorgi & Lionello 2008; IPCC 2013). This surface, thus, formed the amplified drought plot which was compared to natural drought plot (300m$^2$ surface) where trees grew under natural conditions with no rain exclusion." RC2: L93 – 101: This wording was difficult to understand. How did you determine the extent of drought? How do you know this was indeed a drought stress? AC: To answer to this point, we have added a new graph showing the ombrothermic diagrams for the 2 plots used. The drought periods were presented in this graph showing the recurrence and length of drought periods for every years. Drought stress occurs when the temperature line is above the precipitation bars in ombrothermic diagrams (Emberger et al. 1963).

RC2: L95 – 97: "During the first year of experiments (2012), 35 % of natural rain was excluded and, afterward, 33.5 and 35.5 % were excluded (2013 and 2014, respectively) corresponding for three years, to 2 months for natural treatment and 5 months for amplified treatment of drought period." This text should be rewritten to clearly describe the differences between the natural and amplified drought treatments in terms of rainfall exclusion and periods of application, i.e. what 2 month period? What 5 month period? AC: We have rewritten the paragraph to be clearer. Together with the ombrothermic

diagram, we hope that the experimental precipitation exclusion is better explained (lines 93-105).

RC2: Was there any sampling conducted prior to the experiment or during the experiment on non-drought stressed trees for comparison? AC: There was no sampling conducting prior to the experiment on non-drought stress trees. Indeed, we have two treatments: one where trees are submitted to natural rain (and so natural Mediterranean summer drought) and a second one where trees are submitted to amplified drought (more or less 30% according to climatic models) during the tree growth period.

RC2: L229 - 231: Nevertheless, our results suggested that methanol emissions responded strongly to light and temperature during the day whereas, during the night, they responded to temperature. See General Comments for suggested clarification. AC: We better structured the manuscript by introducing: "All six BVOCs monitored showed daytime light and temperature dependencies while three BVOCs (methanol, acetone and formaldehyde) also showed emissions during the night despite the absence of light under constant temperature. Âż in the beginning result section (lines 212-215).

RC2: L261 – 262: "Moreover, some phenomenon, such as the burst in early morning (methanol and acetaldehyde) or the deposition/uptake (formaldehyde), were not modelled by L+T or T algorithm." Figures 4 and S3 show that the models do accurately simulate the emission burst for methanol as well as the formaldehyde deposition, albeit the models both show a slight lag in the hour of the day in just the autumn natural drought conditions. AC: As we said above: "the burst of methanol is observed only in spring and summer between 6am and 8am. For exemple, methanol emission reach almost $0.8\mu gC.gDM-1.h-1$ during the burst contrasting with the previous emission (less than 0.01). The graph shows that none of the model (yellow and dotted lines) fit this burst. Concerning formaldehyde deposition (S4), even if the models follow the same emission pattern than the observations, they don't show any negative emission and so can't accurately allow to estimate the deposition".

Emberger L., Gaussen H., Kasas M. & DePhilippis A. (1963). Carte bioclimatique de la zone méditerranéenne. UNE SC OF AO, Paris, carte et annexes.

Giorgi F. & Lionello P. (2008). Climate change projections for the Mediterranean region. Global and Planetary Change, 63, 90-104.

IPCC (2013). In: Contribution of working group I to the fith assessment report of the intergovernmental panel on climate change. Cambridge Univeristy Press Cambridge.

---

## Author Response (AR2)

**Thanks for your answer.**

**Scientific comments:**

**Page 6, lines 196-197: "The emissions of all BVOCs followed during this experimentation were reduced under amplified drought compared to natural drought, especially in spring and summer (Table 1) except for acetaldehyde emissions". Was this statistically significant? Can you give p-values e.g. in Table 1.**
*We added p-value in table 1.*

**Page 7, line 223: "…a slight underestimation of emissions (sl = 0.84, P < 0.05)…" how can the algorithm, if fitted seasonally to the data, underestimate the emission? If I understand correctly the EF to calculate the emission was obtained by fitting the algorithm to the exactly same dataset. If I misunderstood, others may also and thus this needs to be written more clearly. Also, is the p-value for the correlation or have you tested the deviation from 1:1 line? You should have the latter to be not misleading the readers. These two questions concern also at least the following claims:**
**-Page 7, line 231: "…a slight underestimation was observed (sl = 0.87, P < 0.05)".**
**-Page 7, lines 236-237: "…underestimations were observed in spring and summer (sl = 0.72, and sl = 0.57, P < 0.05, respectively)"**
**-Page 7, line 238: "…underestimation was only observed in summer (sl = 0.80, P < 0.05)".**
**-Page 8, lines 263-264: "…slight underestimations were observed (sl = 0.88, P < 0.05 and sl = 0.69, P < 0.05, respectively)".**
**-Page 8, line 265: "…line overestimation of modelled emissions (sl = 1.27, P < 0.05)".**
*Indeed, EF was determined with the same dataset as used to modelled emissions. EF was the slope of the correlation between emission rates and Cl*Ct for L+T algorithm and the correlation between emission rate and Ct for T algorithm (line 172-176).*
*In our study, temperature, light, season and the water stress were factors taken into account to modeled BVOCs emissions. An under (or –over) estimation between measured and calculated emission rates highlights the fact that the driving parameters considered in the algorithm (temperature, light, season, drought) did not allow to explain 100% of emissions. Thus, it seems that other factors which are not taken into account in our study influenced emissions*
*We tested the deviation of the slope from 1:1 line and the p-value presented for each example cited above was the p-value of this slope comparison test. We detailed more this test in material and methods (line 184-186) as well as in the manuscript.*

**Technical comments:**

**In many locations: "BVOCs emissions" should be "BVOC emissions"**

*We checked the BVOCs or BVOC throughout the manuscript.*

**Page 2, lines 68-70: "These studies reveal that there are still some misunderstandings at the level of emission mechanisms since some works showed increases (Funk et al. 2004; Monson et al. 2007) or decreases of isoprene emissions (Brüggemann & Schnitzler 2002;**

**Fortunati et al. 2008)". Would this indicate rather lack of understanding rather that misunderstanding?**

*We changed that in the manuscript (line 68).*

**Page 3, lines 71-72: "Moreover, the sensitivity of isoprene and highly volatile BVOCs emissions to recurrent water stress (few years) under in situ conditions is clearly missing". Is it sensitivity or rather understanding of the sensitivity that is missing?**

*It is the understanding of sensitivity that is clearly missing. We changed that in the manuscript (line 71).*

**Page 4, line 116: "air generator". Is this zero air generator?**

*No, it is not a zero air generator. It a pump made inside by PTFE. We changed "air generator" into "pump" to avoid confusion (line 116).*

**Page 4, line 132: "Exchanges of CO2 and H2O from the enclosed branches were…" should be "Exchange of CO2 and H2O from the enclosed branches was…"**

*Yes indeed, we changed that in the manuscript (line 132).*

**Page 4, line 144: "catalyzer". What is catalyzer? It has not been described before. Is it air generator mentioned in line 116?**

*No, it is not the same thing. The catalyst consists in a 25 cm long stainless steel tubing, filled with platinum wool and heated at 350°C to efficiently remove VOCs from sample and measure potential instrumental background levels. This has been specified in the manuscript (lines 144-146).*

**Pages 4-5, lines 144-145: "Each sample was analysed every hour, with 15min of analysis". How can this be as there are five samples according to the previous sentence.**

*BVOCs emissions were sampled concomitantly for one tree under natural and amplified drought, during 1 or 2 days (line 115). During the sampling day, there was a measure cycle thanks to the valve system (Vici) which allows us to measure BVOC emissions from each sample line (amplified drought – inlet air – natural drought – ambient air – catalyzer) every hour during 15min. However, we followed 5 trees for each plot (natural and amplified drought) by moving enclosures chambers from trees to trees during two weeks.*

**Page 5, line 148: "A calibration gas standard" Which compounds were in gas standard?**

*It is a mixture of 14 aromatic organic compounds (100 ppb each in nitrogen). It includes benzene, toluene, styrene, m, o, p-xylene, ethylbenzene, 1,2,4 trimethylbenzene 1,3,5 trimethylbenzene, chlorobenzene, 1,2 dichlorobenzene, 1,3 dichlorobenzene, 1,4 dichlorobenzene, and 1,2,4 trichlorobenzene. We added this in the manuscript (line 150-151).*

**Page 5, Equation (1) expresses gas exchange per foliar mass. However, on page 4, lines 138-139 you state "Gas exchange were hence expressed in a leaf surface basis", leading to inconsistency.**

*Gas exchange in terms of $CO_2$ and $H_2O$ was expressed in a leaf surface basis whereas BVOCs emissions were expressed per foliar mass. To avoid misunderstanding, we wrote "Pn and Gw were hence, expressed in a leaf surface basis" instead of "Gas exchange were hence expressed in a leaf surface basis" (line 138).*

**Page 6, lines 212-213: "All six BVOCs monitored showed daytime light and temperature dependencies (isoprene, degradation products of isoprene and acetaldehyde),…" The beginning of this sentence is confusing as it mentions six BVOCs but then only gives three in parentheses.**
*We changed that in the manuscript (line 293-294).*

**Page 7, lines 247-248: "However, some observed phenomena suggested that methanol emissions was sustained by temperature alone at certain moment of the day". This sentence is a bit confusing and vague. Could it better as "However, some observed phenomena suggested that methanol emission was sustained by temperature alone in the absence of light".**
*Yes indeed, we changed this sentence (line 249-250).*

**Table 1: Too many significant numbers in part of the values. E.g. 107.7± 18.6 should be expressed as 110 ± 20 (as the two last digits in the original form contain no information).**
*We changed that in table 1.*

---

## Author Response (AR3)

**The manuscript has improved considerably. I have one detail that I find confusing still in the analysis. In your reply you didn't really answer to my question on it.**

**In your reply you write: "Indeed, EF was determined with the same dataset as used to modelled emissions. EF was the slope of the correlation between emission rates and Cl\*Ct for L+T algorithm and the correlation between emission rate and Ct for T algorithm (line 172-176)".**

**If you determine the EF as the slope between the emissions and Cl\*Ct or Ct, and then plot the data modelled with these EFs against the very same data the EFs were derived with, I would expect 1:1 line, unless non-linear data transformations have been done. If that is not the case I would doubt the fitting procedure.**

*EF was determined as the slope of the correlation between ER (observed emissions) and algorithms values of Cl\*Ct or CT for each tree, each compound, each season and each treatment. After this, we calculated the mean of EF for each condition (for example, formaldehyde in spring under natural drought) with n=5. It allowed us to introduce the inter-individual variability of BVOC emissions. The EF mean was used to calculate the modelled emissions as follows:*

$$modelled\ emissions = EF * ClCt\ (for\ L{+}T\ algorithm)$$

*Or*

$$modelled\ emission = EF * CT\ (for\ T\ algorithm)$$

*We cannot have a 1:1 line nor a R² of 1 because when we determined the EF with the correlation between ER and Cl\*Ct, R² obtained was not equal to 1 because light and temperature were not the only parameters influencing the BVOC emissions.*

**And further: "An under (or –over) estimation between measured and calculated emission rates highlights the fact that the driving parameters considered in the algorithm (temperature, light, season, drought) did not allow to explain 100% of emissions. Thus, it seems that other factors which are not taken into account in our study influenced emissions".**

**This seems to be misinterpretation, as you have used the very same data to create the EFs that you use to evaluate them, and thus the EF implicitly is influenced by all parameters in seasonal timescale. To me it looks like there is something strange in the procedure of obtaining EFs, calculating the emission and then comparing them back to the original data.**

**If we look for example methanol emission in autumn in Figure 5, we see that the L+T algorithm constantly underestimates the emission. This is very strange if the EF used in the calculation is obtained from this same dataset. One would expect part of the data be above, part below the algorithm. Were the algorithms fitted to original data and not the averages shown here? Could the distribution of the data be very skewed to cause this? I would like to have a short explanation on how the systematic deviation of the modeled emission and measured one is possible in this case.**

*For example, we performed the correlation between Cl\*Ct and ER for methanol in summer for only one tree under natural drought. We obtained a slope equal to 0.6242 (graph A) which will be our EF value to calculate modelled emission of methanol in summer under natural drought. ER (experimental emissions rates) and EM (modelled emissions) are presented in graph B. We observe that there is underestimation of methanol emissions from algorithm L+T. Thus, even when modelled emissions are calculated without averaging, we obtained the same results because light and temperature cannot fully explain methanol emissions.*

[Figure]

[Figure]

Such an approach used for calculating of modelled emissions then, compared to observed emissions, has been used in many other publications. They determined EF with observed emissions (ER) and validated their EF with the same data set. To determine EF, there are two methods:

1- Using EF as the slope of the correlation between ER and ClCt values (Tarvainen et al. 2005; Dindorf et al. 2006; Grabmer et al. 2006; Holzke et al. 2006; Harley et al. 2014).

2- Averaging all emissions rates occurring at standard conditions during the measurement (1000±200 μmol.m-2.s-1 and 30±2°C) (Fares et al. 2011).

In our study, we tested both methods but we obtained a better fit between modelled emissions and experimental emissions with the method presented in the article (method 1), because a larger dataset is taken into account to determine the EF in this method.

**Furthermore, you write: "In our study, temperature, light, season and the water stress were factors taken into account to modeled BVOCs emissions". Actually, if I understand correctly, only temperature and light are taken into account explicitly (in algorithm). Drought effect is only implicitly included in the seasonal variation of EF.**
*Indeed, in our study, only light and temperature are taken into account in the algorithms. Water*

*stress and seasonality are taken into account indirectly since we determine an EF for each compounds at each season and each treatment (natural and amplified drought).*

*Dindorf T., Kuhn U., Ganzeveld L., Schebeske G., Ciccioli P., Holzke C., Köble R., Seufert G. & Kesselmeier J. (2006). Significant light and temperature dependent monoterpene emissions from European beech (Fagus sylvatica L.) and their potential impact on the European volatile organic compound budget. Journal of Geophysical Research: Atmospheres, 111.*

*Fares S., Gentner D.R., Park J.-H., Ormeno E., Karlik J. & Goldstein A.H. (2011). Biogenic emissions from< i> Citrus species in California. Atmospheric Environment, 45, 4557-4568.*

*Grabmer W., Kreuzwieser J., Wisthaler A., Cojocariu C., Graus M., Rennenberg H., Steigner D., Steinbrecher R. & Hansel A. (2006). VOC emissions from Norway spruce (Picea abies L.[Karst]) twigs in the field—results of a dynamic enclosure study. Atmospheric Environment, 40, 128-137.*

*Harley P., Eller A., Guenther A. & Monson R.K. (2014). Observations and models of emissions of volatile terpenoid compounds from needles of ponderosa pine trees growing in situ: control by light, temperature and stomatal conductance. Oecologia, 176, 35-55.*

*Holzke C., Hoffmann T., Jaeger L., Koppmann R. & Zimmer W. (2006). Diurnal and seasonal variation of monoterpene and sesquiterpene emissions from Scots pine (Pinus sylvestris L.). Atmospheric Environment, 40, 3174-3185.*

*Tarvainen V., Hakola H., Hellŕn H., Hari P. & Kulmala M. (2005). Temperature and light dependence of the VOC emissions of Scots pine. Atmospheric Chemistry and Physics, 5, 989-998.*

---

## Author Response (AR4)

Dear Saunier et al.,

**Thank you for your detailed reply. Now I understand where the discrepancy between modeled and measured emission originate from. As I said, one would expect modeled the modeled versus measured comparison to follow 1:1 line (of course with R2 below unity) when the emission factor is obtained with the same data that is used for comparison. In your study, you have used only the slope to calculate the fluxes, disregarding the intercept. In the case of your example, methanol in summer, natural drought, this leads to an underestimation of around 0.05 gC gdw-1 h-1. We can also see from Figure 3-5 of the manuscript that the systematic underestimation occurs for those compounds which exhibit considerable night time emission, and thus intercept on fitting. For compounds with zero nigh time emission the algorithm works much better. The proper way of fitting emission algorithms is to force the intercept to zero, or to describe the intercept, i.e. the light independent emission, with an additional parameterization, as done e.g. in the case of monoterpene emission from boreal trees by Ghirardo et al. (2010).**

*We added in the manuscript that emissions factors (EF) were obtained from correlations between experimental emissions rates (ER) and $Cl*Ct$ or $C_T$ without forcing data to pass through the origin (this approach is explained in the manuscript and detailed in the Appendix B with the formula).*

*Moreover, in this corrected version of the manuscript, we show the mean of intercept (called B in our manuscript) under each condition (drought treatment and season) in the tables S1 and S2 of supplementary files. We performed also tests to check if our intercepts were different from 0.*

**You should modify the manuscript to reflect the origin of the underestimation in the night time emission of certain compounds.**

*We added in the manuscript a sentence about this point (lines 283-285).*

---

## Author Response (AR5)

We thank the editor for all the useful comments which help us to improve our MS. We carefully analyzed the new comments, as shown hereafter.

**Your modification to the manuscript does not yet fully satisfy. In matter of fact, if I plot the (visually estimated) points in the first panel of Fig 3, I get slope of 1.06 (close to one, (intercept of -0.80, Matlab polyfit function), not 0.84 as in the manuscript. So, this makes the data processing still look a bit suspicious and the one sentence added to the revised manuscript does not fully cover this.**

We have tried to understand your results (slope = 1.06 against our slope = 0.84) and realized that differences make sense since:

*a)* in our manuscript, the fit between modelled (EM) and measured emission rates (ER) are performed this way : EM = a ER + b (graph a). So, slopes and $R^2$ values shown in Figs 3, 4, 5 and supplementary figures, are calculated from these correlations. According to the example you chose, EM = 0.84 ER + 1.4744.

*b)* your calculations are obtained when ER = a EM + b  (graph b) . So, ER = 1.06 EM - 0.9155. Note that $R^2$ = 0.8951 are the same in a) and b).

So, either way (EM = a ER + b  or ER = a EM + b), the model slightly underestimates the emission rates. However, when plotting ER = a EM + b, this conclusion can only be reached if one accounts for the intercept, which is negative (-0, 9155). If only the slope value is taken into account (1.06), one would conclude that the fit is extremely close to 1. Thanks to your comments, we think that it is very important to clarify that in our manuscript ER was the x axis while EM was the y axis (line 184).

[Figure]

**If I understand correctly, the only difference in correlating emission against CTCL, as you do for obtaining EF, and plotting emission against modeled emission, as you do to obtain slopes in Fig. 3-5, is that in the latter one you multiply the x-axis by EF (as E_model=EF*CTCL). Am I correct in this?**

Indeed. As explicitly written in the manuscript (lines 167 - 169), the EF is calculated under each treatment as the mean of the emission factors for N= 5 trees/treatment. This approach allows to take into account the variability of emissions observed during our experiment.

**Thus, in the case when you have no nigh-time emission (intercept = 0) the slope between modeled emission and measured emission should be close to unity. However, in several cases this is not the case (Fig 3, spring natural drought; Fig 4, several of the plots). Have you looked at how similar are the emission against CTCL plots to the emission against model emission plots? Could these be added as supplementary material?**

**For isoprene (Fig. 3):**

In spring and under natural drought, $R^2$ of the correlation between measured emissions and $C_l*C_t$ varies from 0.77 to 0.88 whereas, in summer when no underestimation was observed, $R^2$ is a bit better and varies from 0.73 from 0.97 which can explain the discrepancies between modeled emissions and measured emissions observed in spring.

**For acetaldehyde (Fig. 4):**

On the one hand, the burst of acetaldehyde in the early morning (Fig 4) is not explained by any algorithm ($C_l*C_t$ or $C_T$), which can explain that slope is different from 1.

On the other hand, it is important to highlight that some trees showed weak correlations between their emissions and $C_l*C_t$ (or $C_T$), although these correlations were significant. For instance, in summer and under natural drought, $R^2$ of the correlations between $C_l*C_t$ and ER varies from 0.34 to 0.90 among the five trees studied (Table S6, new supplementary file), which can explain that the model did not fit well to measured emissions. These relatively weak correlations suggest that light and temperature are not the only factors driving acetaldehyde emissions in all trees and that there were large differences between trees. The discussion considers now this issue (see lines 242 - 247).

In order to highlight that tree BVOC emissions do respond differently to light and temperature (or temperature alone), the new version of the manuscript shows 6 new tables in supplementary file showing $R^2$ and p-value of correlations between $C_l*C_t$ (or $C_T$) and experimental emissions of each compound, tree by tree (tables S1 to S6). This data is separated by seasons and treatments. To our knowledge, this type of information clearly misses in past studies while it is indeed very relevant since it reflects the natural variability among trees growing under the same natural conditions.

**Because of the above mentioned possible discrepancy in the data processing, I feel the following sentence, "The modelled emissions were very representative of measured emissions except in spring for natural drought when we obtained a slight underestimation of emissions (sl = 0.84, P < 0.05) maybe, because light and temperature, in spring, were not the only parameters driving isoprene emissions", may be over-interpretation.**

According to the editor comments, we changed this sentence in the manuscript as follows:

Modeled emissions were roughly very representative of measured emissions. We note however that in spring, under natural drought, emissions were slightly underestimated (sl = 0.84, $R^2$=0.90, P < 0.05). This result suggests that although light and temperature remain the main factors driving isoprene emissions in spring, other parameters explain 10 % of isoprene emission variability. In spring, plants likely require to produce more isoprene to protect the establishment of the photosynthetic machinery in the new leaves which could slightly modify the effects of light and temperature on isoprene emissions (lines 222 – 227).

**Page 8, lines 283-285: "Predicting emissions rates of these 3 compounds (methanol, acetone and formaldehyde), during the night, seem to require other parameters such as a temperature threshold, below which methanol, acetone and formaldehyde synthesis and so emissions do not occur (Ghirardo et al. 2010)". You should remove citation Ghirardo et al. from this as that paper does not deal with methanol, acetone or formaldehyde.**

As suggested, we removed this reference from the manuscript.

---

## Author Response (AR6)

Thanks for all your comments.

From you answers and manuscript I understand now that you have obtained the EF as slope of the CTCL or CT (on x axis) against measured emission (y axis) without forcing line through origin. Then you compare the algorithm performance by slope of measured emission (x axis) against modeled emission (y axis). In the case of non-linear relations this will lead to slopes below 1, even though modelled emission (x axis) vs measured emission (y axis) would have slope of unity. Thus, the result concerning the slopes being below unity may be partly due to non-linear relation of modeled and measured emission. The nonlinearity of relation for isoprene is actually interesting, as it can be due to light penetration deeper to canopy in high light levels. The algorithm used here is anyway a big leaf approximation.

**Page 1, lines 22-13: "these 3 last compounds detected under the same ion". Actually, not the same ion but same mass-to-charge ratio.**
We changed that in the manuscript as: these 3 last compounds detected under the same m/z (lines 23 and 150)

**Page 2, lines 69-70: "…impact of water stress on highly BVOCs emissions." What does "highly" mean here? Please reformulate.**
In this sentence, we talked about highly volatile BVOC emissions. So, we changed that in the manuscript as: impact of water stress on highly volatile BVOCs emissions (e.g. methanol) (line 70).

**Pages 5-6, lines 181-182: The slope of those correlations indicate if there was an under- or over- estimation of modelled emissions when sl < 1 and sl > 1, respectively. There is also under/over estimation if significant intercept, even if slope is 1. See fig 5, autumn methanol AD.**
We changed this sentence as : these correlations indicate if there was an under- or over-estimation of modelled emissions with sl < 1 and sl > 1, respectively, or if the intercept (called "b" afterwards) are different from 0. For that, slope comparison tests were performed to check for slope significant differences from 1 and intercept tests were performed to check for intercept significant differences from 0 (line 182 – 187).